# Netcombin: An algorithm for constructing optimal phylogenetic network from rooted triplets

Hadi Poormohammadi[1,2]*, Mohsen Sardari Zarchi[1]

**1** Department of Computer Engineering, Meybod University, Meybod, Iran, **2** School of Biological Sciences, Institute for Research in Fundamental Sciences (IPM), Tehran, Iran

* pormohamadi@meybod.ac.ir

**Data Availability Statement:** All relevant data are within the manuscript and its Supporting Information files.

## Abstract

Phylogenetic networks construction is one the most important challenge in phylogenetics. These networks can present complex non-treelike events such as gene flow, horizontal gene transfers, recombination or hybridizations. Among phylogenetic networks, rooted structures are commonly used to represent the evolutionary history of a species set, explicitly. Triplets are well known input for constructing the rooted networks. Obtaining an optimal rooted network that contains all given triplets is main problem in network construction. The optimality criteria include minimizing the level or the number of reticulation nodes. The complexity of this problem is known to be NP-hard. In this research, a new algorithm called Netcombin is introduced to construct approximately an optimal network which is consistent with input triplets. The innovation of this algorithm is based on binarization and expanding processes. The binarization process innovatively uses a measure to construct a binary rooted tree $T$ consistent with the approximately maximum number of input triplets. Then $T$ is expanded using a heuristic function by adding minimum number of edges to obtain final network with the approximately minimum number of reticulation nodes. In order to evaluate the proposed algorithm, Netcombin is compared with four state of the art algorithms, RPNCH, NCHB, TripNet, and SIMPLISTIC. The experimental results on simulated data obtained from biologically generated sequences data indicate that by considering the trade-off between speed and precision, the Netcombin outperforms the others.

## Introduction

Phylogenetics is a branch of bioinformatics that studies and models the evolutionary relations between a set of species or organisms (formally called taxa) [1, 2]. The tree structure is the basic model which can show the history of tree-like events such as mutation, insertion and deletion appropriately [1–5]. The main disadvantage of the tree model is its disability to show non-treelike events (more abstractly, reticulate events) like recombination, hybridization and horizontal gene transfer [2, 6]. To overcome this weakness, phylogenetic networks are introduced to generalize phylogenetic trees and represent reticulate events [1, 2, 7–13].

**Funding:** This research is partially supported by a grant from Institute for research in fundamental sciences (IPM), Tehran, Iran by a grant number BS-1396-01-06. There is no additional external funding received for this study.

**Competing interests:** The authors have declared that no competing interests exist.

The structures of trees and networks can be divided into two groups, rooted and un-rooted. Rooted structures can show reticulate events, explicitly. Hence this structure has received more attention recently for constructing networks. The rooted structures are always rooted trees or rooted networks. These structures contain a unique vertex called root with in-degree 0 and out-degree at least 2 [1, 2, 7]. Fig 1a shows an example of a rooted tree.

Usually rooted structures are represented in the binary form, i.e. the out-degree of each vertex is at most 2. In a rooted binary tree, the out-degree of all vertices except the leaves are 2, the out-degree of leaves are 0, the in-degree of all vertices except the root, is 1 (Fig 1b). Formally in rooted structures a common ancestor for a given set of taxa is considered as root [1, 2, 7].

One famous approach to build phylogenetic networks is constructing them up from small trees or networks [1, 2]. Triplet is the smallest tree structure which shows the evolutionary relation between three taxa. The symbol $ij|k$ is used to show a triplet $t$ with $i, j$ in one side of $t$ and $k$ in another side of $t$ (Fig 2) [2, 7].

Triplets are commonly considered as a standard input for building rooted structures [2, 7]. These small tree structures are usually obtained from a set of biological sequence data using standard methods such as Maximum Likelihood (ML) and Maximum Parsmiony (MP) [5, 6]. Also in some cases, the output of some biological experiments is directly in the form of triplets [14]. Moreover, in some experiments, triplets are generated randomly to evaluate a model [5].

A network is called level-$k$ if the maximum number of reticulation nodes (nodes with in-degree two and out-degree 1) in each its biconnected components is $k$. [15] (Fig 4b). The optimal network is defined based on the two optimality criterions i.e. minimizing the number of reticulation nodes or the level of the final network [2, 6, 7]. For a given set of triplets as input, the main challenge is to construct an optimal rooted structure (tree or network) which contains all triplets or equivalently, all triplets are consistent with the obtained structure [2, 6, 7, 12]. In other word, all of the input triplets have to be consistent with the output network and that either the level or the number of reticulation nodes in the output network is to be minimized. Formally, a triplet is consistent with a rooted structure when the triplet is a subgraph of that structure [2, 5–7, 12]. The majority preference is to obtain a rooted tree structure. However, as mentioned before, the tree structure can't represent reticulate events and eventually the reticulate events can't be covered. So, in this case the network construction should be considered.

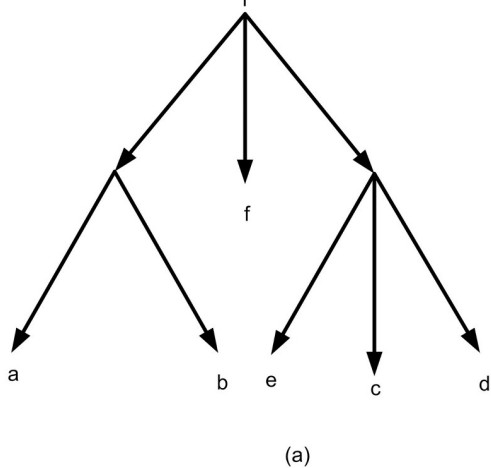

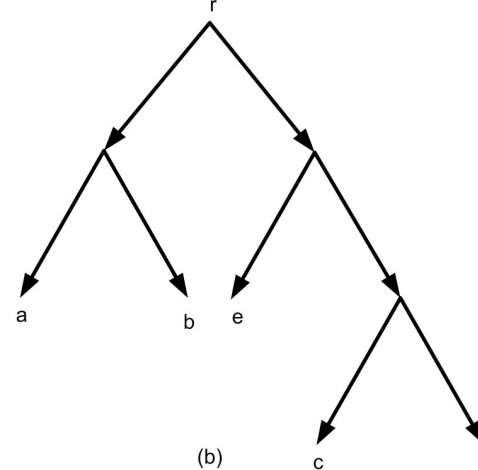

**Fig 1.** (a) A rooted tree in which the out-degree of the root is 3. (b) A rooted binary tree.

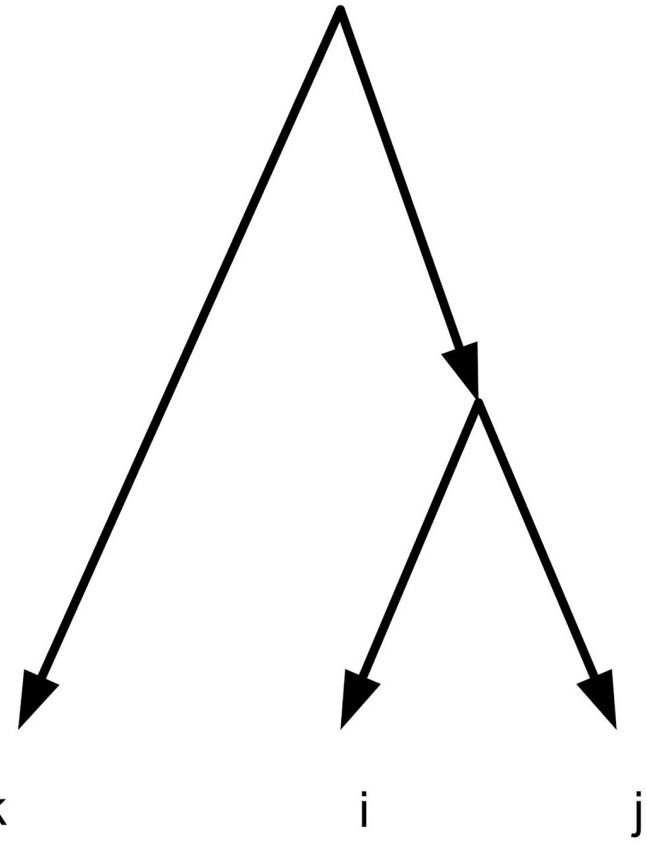

**Fig 2. A triplet _ij|k_.**

In order to build a rooted network from a set of triplets, several algorithms were introduced recently [2, 6–8, 12, 13, 16]. The well-known algorithms are TripNet [7], SIMPLISTIC [6], NCHB [16] and RPNCH [8]. These algorithms find a semi-optimal rooted phylogenetic network that is consistent with a given set of triplets. Because of using heuristic algorithms the result is not necessarily exact optimal. It means that the resulting network is near to optimal which is called semi-optimal. Formally, a rooted phylogenetic network $N$ (network for short) is a directed acyclic graph (DAG) (Fig 3) that is connected and each vertex satisfies one of the following four categories: (i) A unique root node with in-degree 0 and out-degree 2. (ii) Tree nodes with in-degree 1 and out-degree 2. (iii) Reticulation nodes with in-degree 2 and out-degree 1. (iv) Leaves with in-degree 1 and out-degree 0 (Fig 4a). A network is called a network on $X$ if the set of its leaves is $X$. For example the network of Fig 4a is a network on $X = \{l_1, l_2, \ldots, l_7\}$.

Generally the problem of constructing an optimal rooted phylogenetic network consistent with a given set of triplets is known to be NP-hard [17, 18]. When the structure of the input triplets is dense, this problem can be solved in polynomial time order [18]. A set of triplets $\tau$ is called dense if for each subset of three taxa there is at least one information in the set of input triplets [7, 18]. More precisely, a set of triplets $\tau$ is called dense if for a given set of taxa $X$ and each subset of three taxa $\{i, j, k\}$ one of the triplets $ij|k$ or $ik|j$ or $jk|i$ belongs to $\tau$ [7, 18]. For example for a given set of taxa $X = \{a, b, c, d, e\}$, the set of triplets $\tau = \{ab|c, ad|b, be|a, ac|d, ae|c, de|a, bd|c, bc|e, be|d, de|c\}$ is dense.

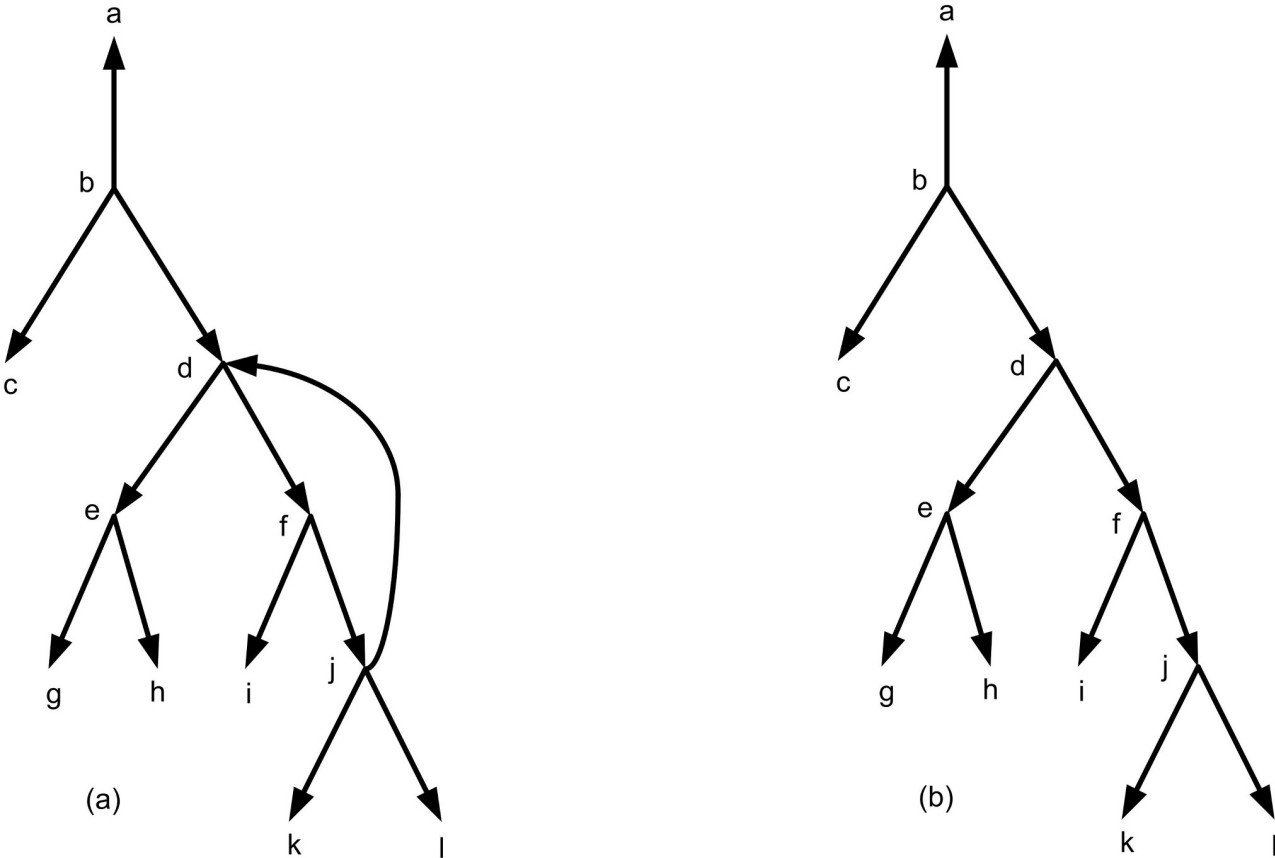

**Fig 3.** (a) A non-acyclic directed graph. (b) A directed acyclic graph (DAG) that is obtained from Fig 3a and by removing the edge (*j, d*).

As mentioned above, density is a critical constraint concerning with constructing a rooted phylogenetic network that contains all given triplets. However, usually there is no constraint on the input triplets and in most cases the input triplets might not be dense. So, introducing efficient heuristic methods to solve this problem is necessary. The desirable goal is to construct a rooted network with no reticulate events i.e. a rooted tree structure. BUILD is the algorithm that was introduced for obtaining a tree structure from a given set of triplets if such a tree exists [19]. In fact, BUILD algorithm decides in polynomial time order if there is a rooted phylogenetic tree that contains all given triplets and then produces an output if such a tree exists. Fig 5, indicates an example of BULID algorithm steps for the given $\tau = \{cd|b, cd|a, cd|e, cd|f, ef|a, ef|b, ef|c, ef|d, db|a, db|e, db|f, da|e, da|f, cb|a, cb|e, cb|f, ab|e, ab|f, ac|e, ac|f\}$.

In tree construction process for a given $\tau$ if BUILD stops, it means that there is no tree structure for the given set of triplets. Fig 6 shows an example for the set of triplets $\tau = \{bc|a, bd|a, cd|a, bc|d, cd|b\}$ in which BUILD algorithm stops. In this case, the main goal is to construct a network structure similar to a tree as much as possible. In other words, constructing a rooted phylogenetic network with the minimum reticulate events is the main challenge.

The simplest possible non-treelike structure (network structure) is level-1 rooted phylogenetic network which also known as galled tree [20]. Fig 7 shows an example of a galled tree. If level-1 networks can not represent all input triplets, more complex (higher level) networks are considered to achieve consistency. LEV1ATHAN is a well-known algorithm to construct

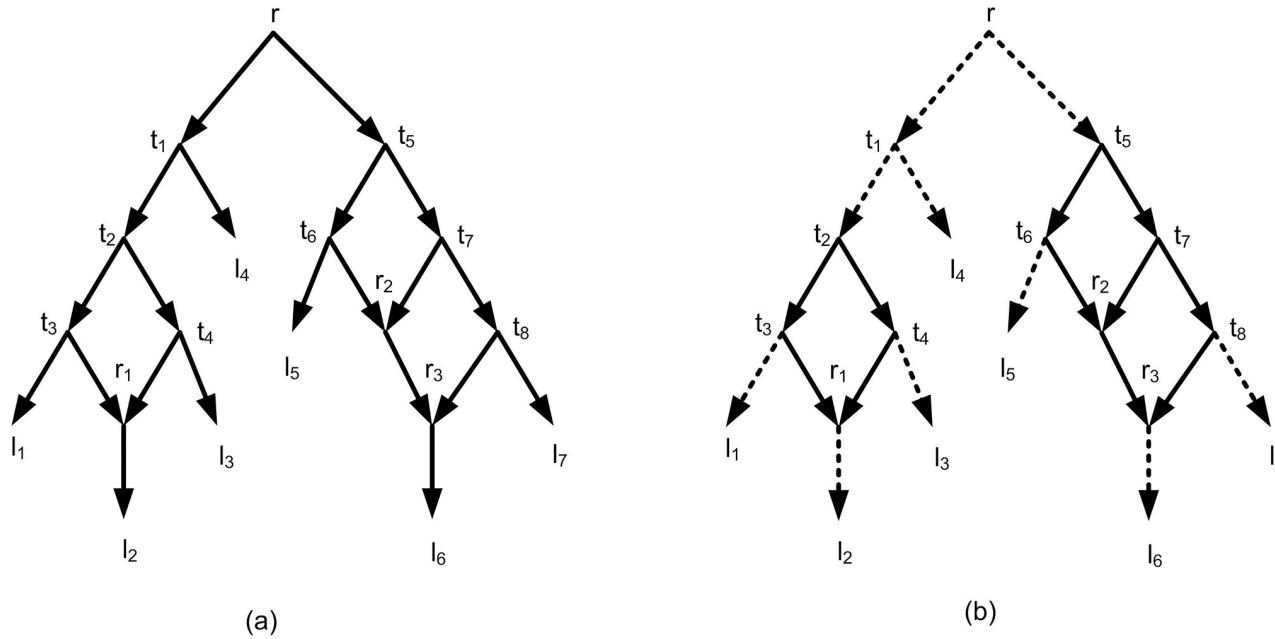

**Fig 4.** (a) A rooted phylogenetic network. $r$ is root, $t_1, t_2, \ldots, t_8$ are tree nodes, $r_1, r_2, r_3$ are reticulation nodes, and $l_1, l_2, \ldots, l_7$ are leaves. (b) The network of Fig 4a has two biconnected components. One of the biconnected component contains $r_1$ and the other contains $r_2$ and $r_3$ and. So the network is a level-2 network.

level-1 networks [21]. In [6] an algorithm is introduced that produces at most a level-2 network (Fig 4).

When more complex networks are needed (e.g. Fig 8), not restricted algorithms such as NCHB, TripNet, RPNCH and SIMPLISTIC are applicable which try to construct a consistent network with the optimality criterions (the level or the number of reticulation nodes) [6–8, 16]. Among the above four mentioned algorithms, SIMPLISTIC is not exact and just works for dense sets of triplets while for the other three methods there is no constraint on the input triplets. This is one of the SIMPLISTIC disadvantages. Moreover for complex networks SIMPLISTIC is very time consuming and has not the ability to return an output in an appropriate time [7].

TripNet has three speed options: slow, normal, and fast. The slow option returns a network near to an optimal network. Normal option works faster compared to slow option, but its network is more complex compared to slow option. Note that slow and normal options return an output in an appropriate time for input triplets consistent with simple and low level networks. However these two options are not appropriate for large data, because by increasing the number of taxa, the set of triplets corresponds to them are consistent with high level networks. Fast option usually output a network in an appropriate time but its network is more complex compared to the two other options. This option is used when the slow and normal options have not the ability to return a network in an appropriate time. It means that fast option just try to output a network and does not consider the optimality criterions. In summary, TripNet has not the ability to return an optimal network in an appropriate time, when input data is large [7]. NCHB is an improvement of TripNet which tries to improve the complexity of the TripNet networks but like TripNet it has not the ability to return an optimal network in an appropriate time for large data [16].

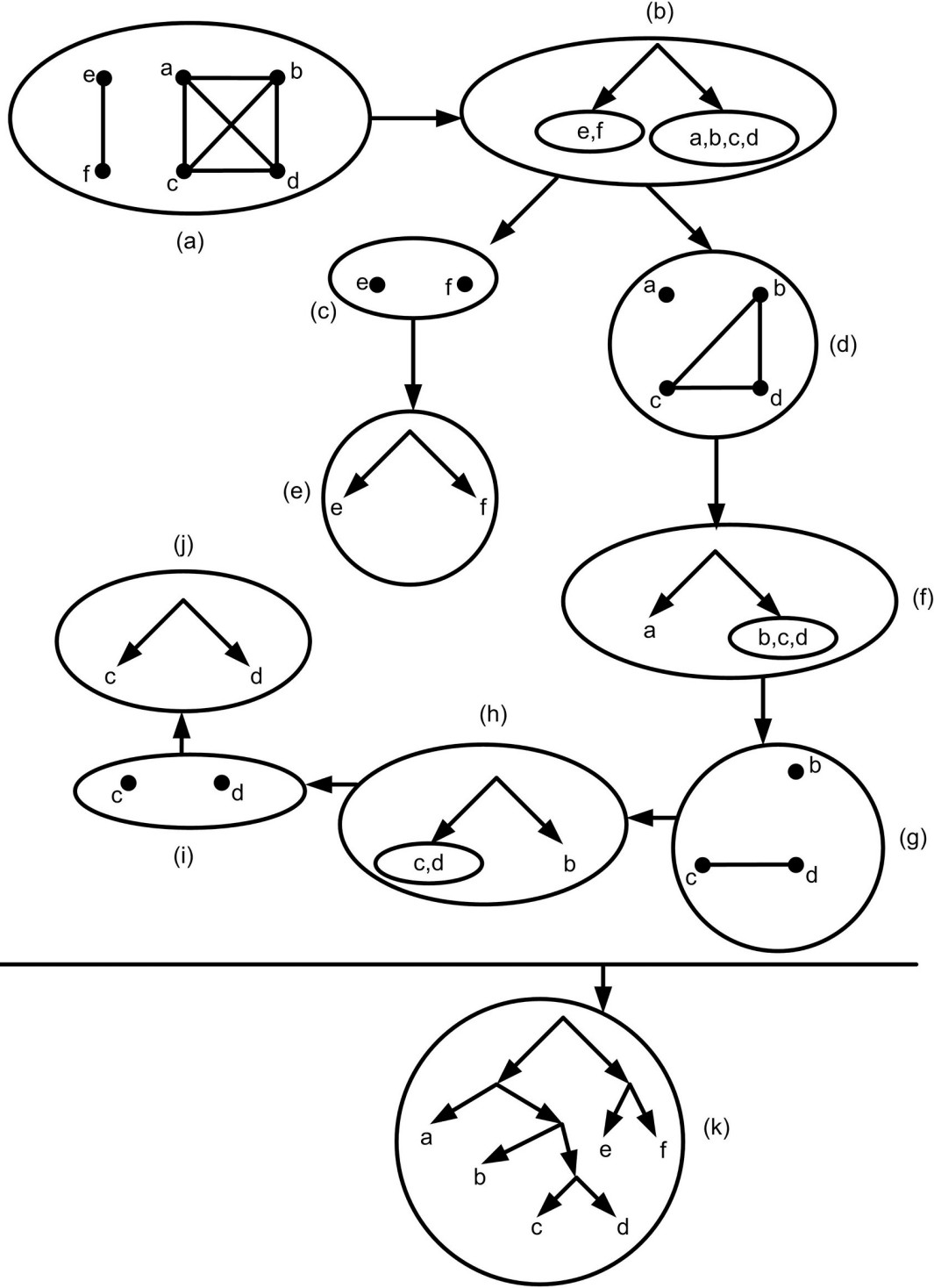

**Fig 5. The Aho graph $AG(\tau)$ is defined based on $\tau$.** The set of nodes are $X = L(\tau) = \{a, b, c, d, e, f\}$ and two nodes $i, j \in X$ are adjacent iff there is a node $x \in X$ such that $ij|x \in \tau$. Also $AG(\tau|A)$ in which $A \subseteq X$ is defined in a similar way. Note that in this way the induced graph of $AG(\tau)$ on the set of nodes $A \subseteq X$ is considered i.e. the set of nodes are $A$ and $i, j \in A$ are adjacent iff there is a node $x \in A$ such that $ij|x \in \tau$. (a) $AG(\tau)$. (b) Based on $AG(\tau)$ the resulting tree is obtained. (c) $AG(\tau|e, f)$. (d) $AG(\tau|a, b, c, d)$. (e) Based on $AG(\tau|e, f)$ the resulting tree is obtained. (f) Based on $AG(\tau|a, b, c, d)$ the resulting tree is obtained. (g) $AG(\tau|b, c, d)$. (h) Based on $AG(\tau|b, c, d)$ the resulting tree is obtained. (i) $AG(\tau|c, d)$. (j) Based on $AG(\tau|c, d)$ the resulting tree is obtained. (k) The final tree consistent with given $\tau$ that is obtained from BUILD algorithm by reversing its steps.

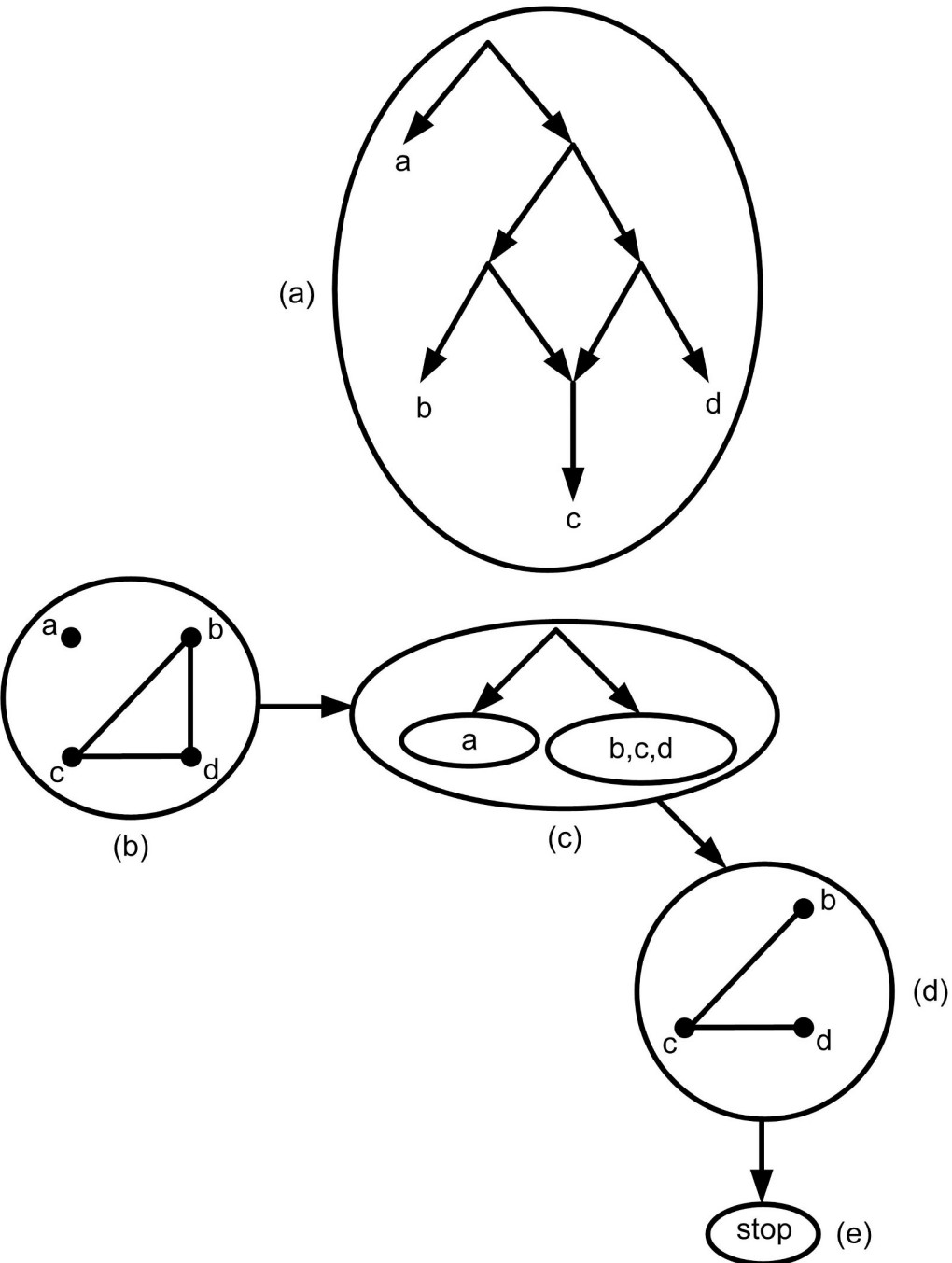

**Fig 6.** (a) The simplest possible network consistent with the given $\tau = \{bc|a, bd|a, cd|a, bc|d, cd|b\}$. (b) $AG(\tau)$. (c) Based on $AG(\tau)$ the resulting tree is obtained. (d) $AG(\tau|b, c, d)$. The graph is still connected. (e) BUILD algorithm stops based on step d.

RPNCH is a fast method for constructing a network consistent with a given set of triplets, but its output is usually more complex considering the two optimality criterions compared to SIMPLISTIC and TripNet networks. It means that although RPNCH is fast but on average, the RPNCH networks are far away from the optimality criterions [8].

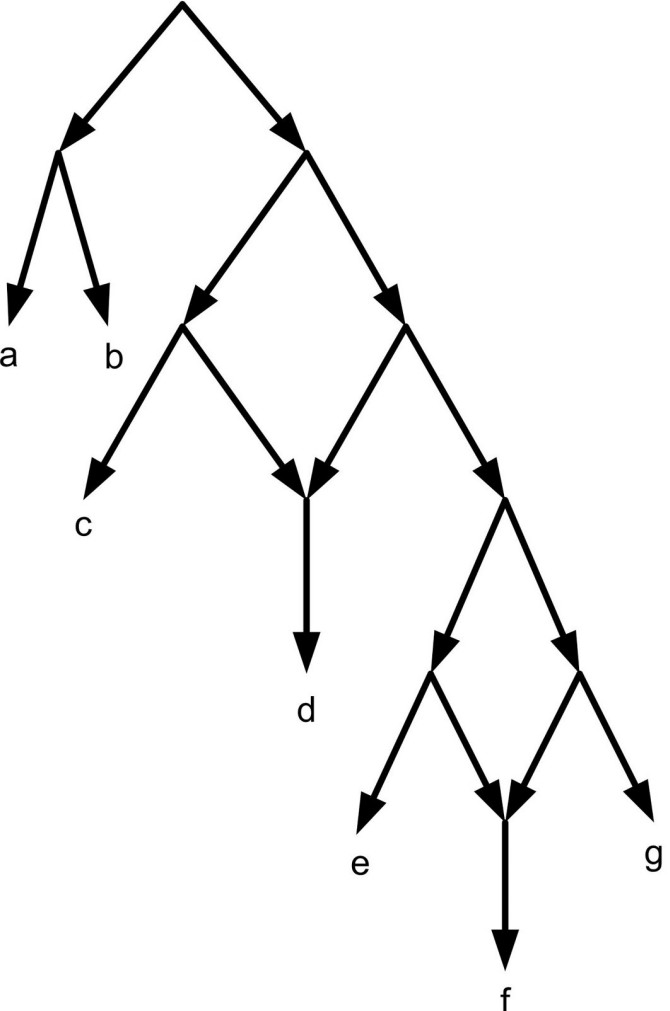

**Fig 7. A level-1 network (galled tree) with two reticulation nodes.**

Generally none of the above four methods have the ability to return a network near to an optimal network consistent with a given set of input triplets in an appropriate time. So the focus of this paper is to introduce a new method called Netcombin (**Net**work **co**nstruction **m**ethod based on **bin**arization) for constructing a semi-optimal (near to optimal) network in an appropriate time without any constraint on the input triplets. In this research our innovation is based on the binarization and expanding processes. In the binarization process nine measures are used innovatively to construct binary rooted trees consistent with the maximum number of input triplets. These measures are computed based on the structure of the tree and the relation between input triplets. In the expansion process which converts obtained binary tree into a consistent network an intellectual process is used. In this process minimum number of edges are added heuristically to obtain the final network with the minimum number of reticulation nodes.

The structure of this paper is as follow. Section 2, presents the basic notations and definitions. In section 3, our proposed algorithm (Netcombin) is introduced and Netcombin time complexity is investigated. In section 4, the new introduced algorithm is compared with

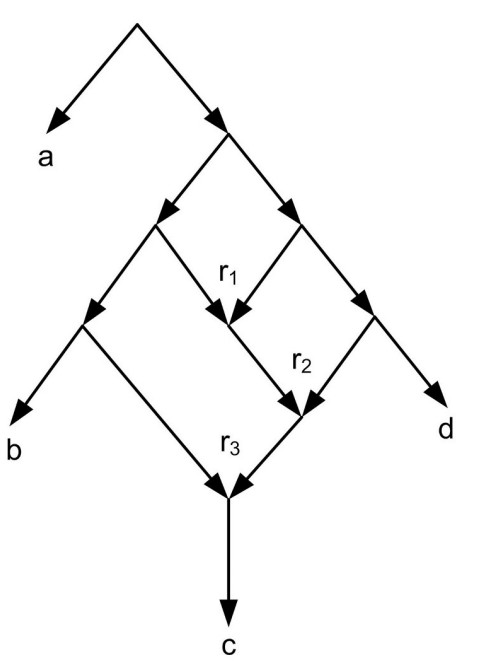
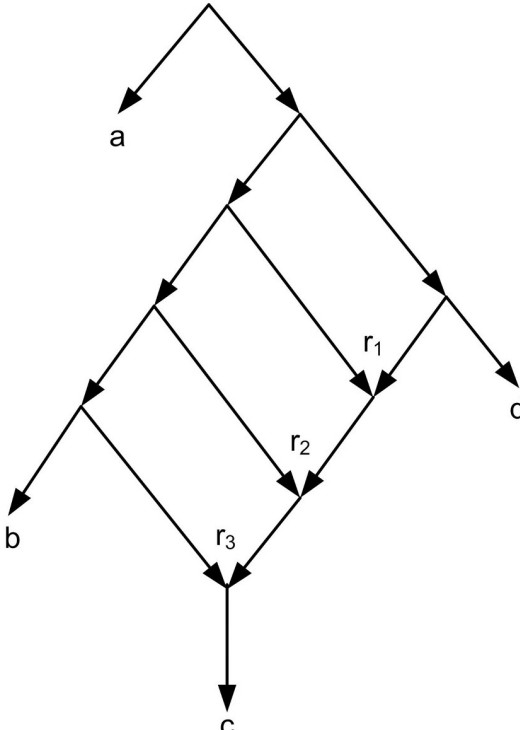

**Fig 8. Two different level-3 networks with three reticulation nodes $r_1$, $r_2$, $r_3$.**

NCHB, TripNet, RPNCH, and SIMPLISTIC and the results are presented. Finally in section 5, the experimental results are discussed.

## Definitions and notations

In this section the basic definitions that are used in the proposed algorithm, are presented formally. From here, a set of triplets and a network are indicated by $\tau$ and $N$, respectively.

A rooted phylogenetic tree (tree for short) on a given set of taxa $X$ is a rooted directed tree that contains a unique node $r$ (root) with in-degree zero and out-degree at least 2. In a tree, leaves are with in-degree 1 and out-degree 0 and are distinctly labeled by $X$. Also inner nodes i.e. nodes except root and leaves, has in-degree 1 and out-degree at least 2 [2, 7]. Fig 1, indicates an example of a tree on $X = \{a, b, c, d, e, f\}$.

The symbol $L_N$ denotes the set of all leaf labels of $N$. $N$ is a network on $X$ if $L_N = X$. A triplet $ij|k$ is consistent with $N$ or equivalently $N$ is consistent with $ij|k$ if $\{i, j, k\} \subseteq L_N$ and $N$ contains two distinct nodes $u$ and $v$ and pairwise internally node-disjoint paths $u \rightarrow i$, $u \rightarrow j$, $v \rightarrow u$, and $v \rightarrow k$. For example, Fig 9 shows that triplets $ij|k$ and $jk|i$ are consistent with the given network, but $ik|j$ is not consistent. A set of triplets $\tau$ is consistent with a network $N$ (or equivalently $N$ is consistent with $\tau$) if all the triplets in $\tau$ are consistent with $N$. $\tau(N)$ denotes the set of all triplets that are consistent with $N$. Let $L(\tau) = \cup_{t \in \tau} L_t$. $\tau$ is a set of triplets on $X$ if $L(\tau) = X$ [7].

Binarization is a basic concept, defined as follows. Let $T$ be a rooted tree and $x$ be a node with $x_1, x_2, \ldots, x_k, k \geq 3$ childeren. These $k$ children are partitioned into two disjoint subsets $X_l$ and $X_r$. Let $X_l = \{x'_1, x'_2, \ldots, x'_i\}$ and $X_r = \{x'_{i+1}, \ldots, x'_k\}$ in which $x'_1, x'_2, \ldots, x'_k$ is an arbitrary relabeling of $x_1, x_2, \ldots, x_k$. If $|X_l| > 1$ then create a new node $x_l$, remove the edges $(x, x'_1), (x, x'_2), \ldots, (x, x'_i)$ and create the edges $(x, x_l)$ and $(x_l, x'_1), (x_l, x'_2), \ldots, (x_l, x'_i)$. Do the

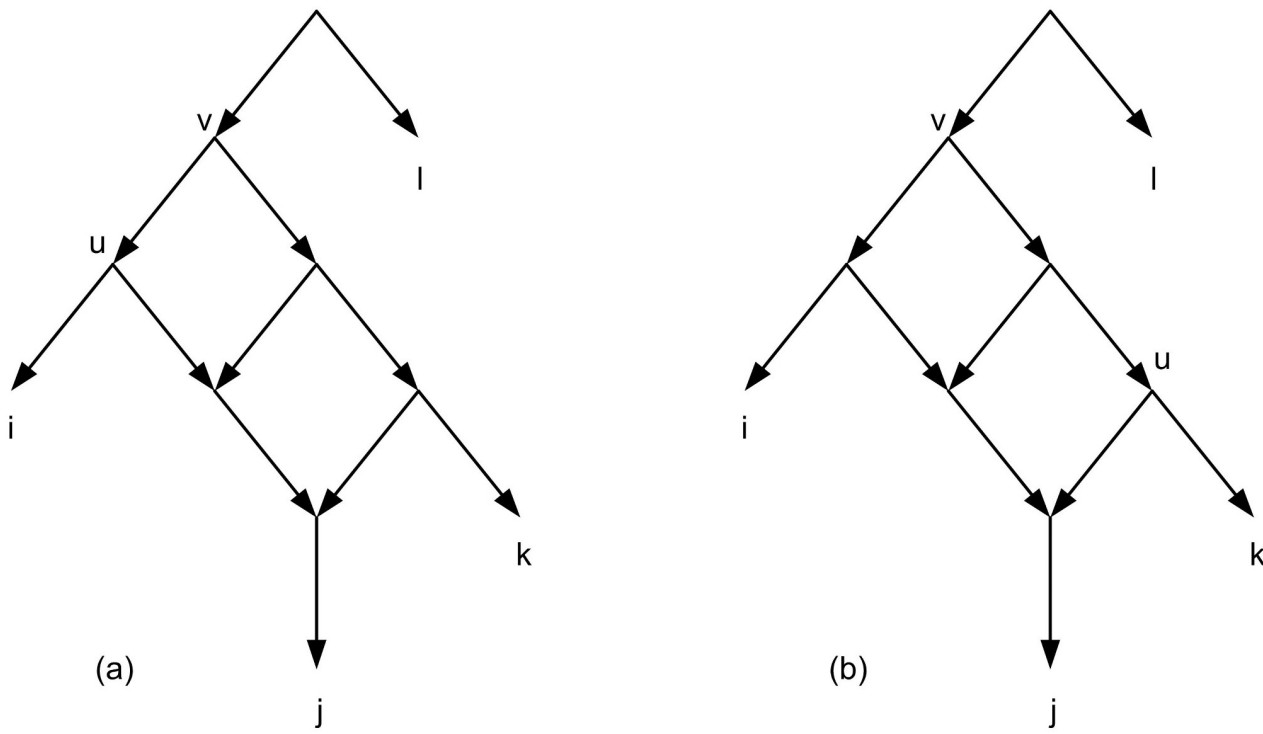

**Fig 9.** (a) Triplet $ij|k$ is consistent with the network. (b) Triplet $jk|i$ is consistent with the network.

same process if $|X_r| > 1$. Continue the process until the out-degree of all nodes except the leaves, be 2. The new tree is called a binarization of $T$. Fig 10 shows an example of a non-binary tree and two samples of its binarizations. Note that there are also more binarizations for the tree which two of them are illustrated. If $T_2$ is a binarization of $T_1$ then $\tau(T_1) \subseteq \tau(T_2)$ [7].

$G_\tau$ the directed graph related to $\tau$, is defined by $V(G_\tau) = \{\{i, j\}: i, j \in L(\tau), i \neq j\}$ ($(i, j)$ is denoted by $ij$ for short) and $E(G_\tau) = \{(ij, ik): ij|k \in \tau\} \cup \{(ij, jk): ij|k \in \tau\}$ [7]. (E.g. Figs 13b and 15b). The height function of a tree and network is defined as follows. Let $\begin{pmatrix} X \\ 2 \end{pmatrix}$ denotes the set of all subsets of $X$ of size 2. A function $h : \begin{pmatrix} X \\ 2 \end{pmatrix} \to N$ is called a height function on X [7]. let $T$ be a tree on $X$, with the root $r$, $c_{ij}$ be the lowest common ancestor of $i, j \in X$, and $l_T$ denotes the length of the longest directed path in $T$. Let $x, y$ be two arbitrary nodes of $T$. $d_T(x, y)$ is the edge path length between $x$ and $y$. For any two $i, j \in X$ the height function of $T$, $h_T$ is defined by $h_T(i, j) = l_T - d_T(r, c_{ij})$. For example, Fig 11.

Let $G_\tau$ be a DAG and $l_{G_\tau}$ be the length of the longest directed path in $G_\tau$. Assign $l_{G_\tau} + 1$ to the nodes with out-degree = 0 and remove them. Assign $l_{G_\tau}$ to the nodes with out-degree = 0 in the resulting graph and continue this procedure until all nodes are removed. Define $h_{G_\tau}(a, b)$, $a, b \in L(\tau)$ and $a \neq b$ as the value that is assigned to the node $ab \in V(G_\tau)$ and call it the height function related to $G_\tau$ [7]. For example Fig 13a to 13d. If $\tau$ is consistent with a tree then $G_\tau$ is a DAG and $h_{G_\tau}$ is well defined [7].

Let $r$ be the root of a given network $N$ and $l_N$ be the length of the longest directed path in $N$. For each node $a$ let $d(r, a)$ be the length of the longest directed path from $r$ to $a$. For any two

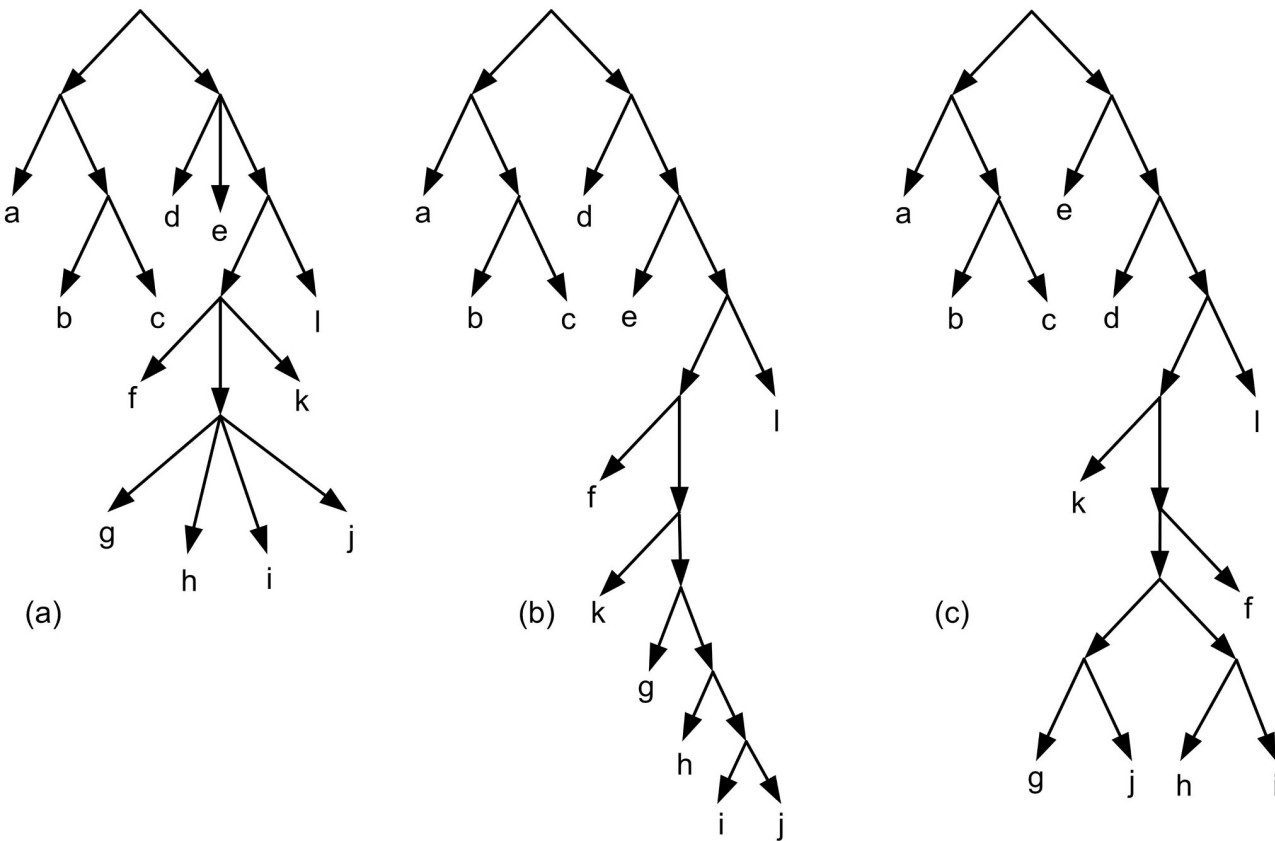

**Fig 10.** (a) A non-binary tree. (b,c) Two samples of binarizations of Fig 10a.

nodes $a$ and $b$, $a$ is an ancestor of $b$ if there is a directed path from $a$ to $b$. In this case $b$ is lower than $a$. For any two nodes $a, b \in L_N$ a node $c$ is called a lowest common ancestor of $a$ and $b$ if $c$ is a common ancestor of $a$ and $b$ and there is no common ancestor of $a$ and $b$ lower than $c$. For any two $a, b \in L(N)$, $a \neq b$, let $C_{ab}$ denotes the set of all lowest common ancestor of $a$ and $b$. For each $a, b \in L(N)$, define $h_N(a, b) = min\{l_N - d(r, c) : c \in C_{ab}\}$ and call it the height function of $N$ [7]. For example for the network of Fig 6a, $l_N = 4$ and $h_N(a, b) = h_N(a, c) = h_N(a, d) = 4$, $h_N(b, d) = 3$, and $h_N(b, c) = h_N(c, d) = 2$.

A quartet is an un-rooted binary tree with four leaves. The symbol $ij|kl$ is used to show a quartet in which $i$, $j$ and $k$, $l$ are its two pairs. Each quartet contains a unique edge for which two its endpoints are not leaves. This edge is called the inner edge of the quartet (See Fig 12) [7].

## Method

In order to build a network $N$ consistent with a given set of triplets $\tau$, the height function $h_N$ related to $\tau$ is defined [7]. The height function is a measure that is used to obtain a basic structure of the final network ($N$) [7]. This basic structure is in the form of a rooted tree. The height function enforce that the obtained rooted tree be consistent with approximately maximum number of triplets of $\tau$. In this research firstly for a given $\tau$, Netcombin assigns a height function $h$ on $L(\tau)$. Then 3 not necessarily binary trees are constructed based on $h$. In the following 9 binarizations of each constructed tree are obtained (i.e. totally 27 binary trees are obtained).

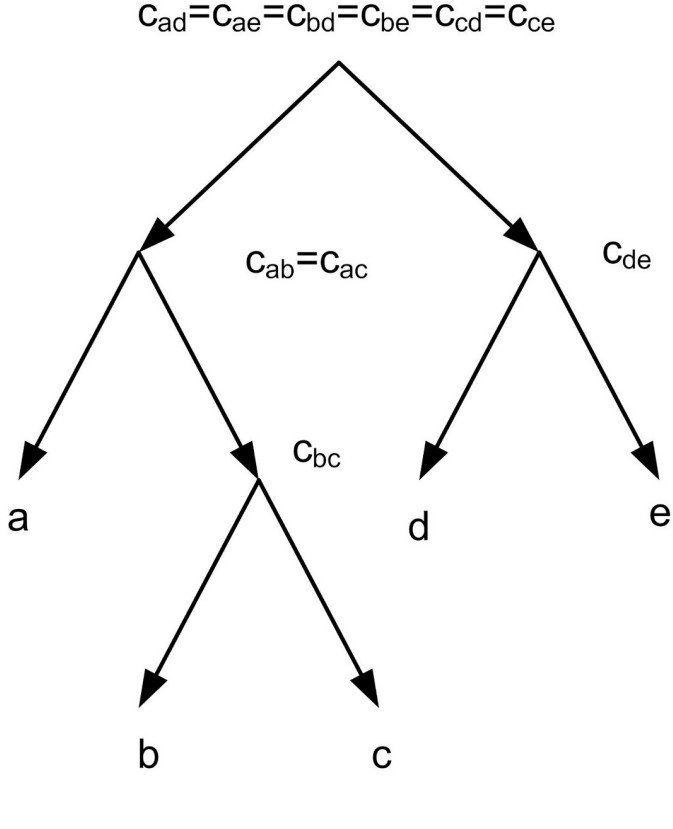

**Fig 11. For the given $T$, $l_T = 3$.** $h_T(a, d) = h_T(a, e) = h_T(b, d) = h_T(b, e) = h_T(c, d) = h_T(c, e) = 3$, $h_T(a, c) = h_T(a, b) = h_T(d, e) = 2$, $h_T(b, c) = 1$.

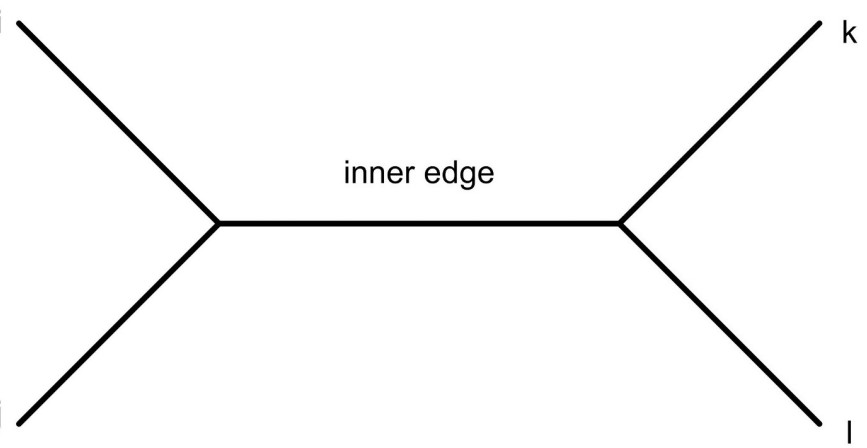

**Fig 12. A quartet $ij|kl$ with the leaves $\{i, j, k, l\}$.**

Finally 27 networks consistent with given $\tau$ are obtained by adding some edges to each 27 binary trees and the optimal network is reported as output as follow: [7]

## Assigning height Function

Let $T$ be a tree with its unique height function $h_T$ and $i, j \in L_T$. The triplet $ij|k$ is consistent with $T$ iff $h_T(i, j) < h_T(i, k)$ or $h_T(i, j) < h_T(j, k)$ [7]. Moreover for a given network $N$ and $i, j, k \in L_N$ with the height function $h_N$ if $h_N(i, j) < h_N(i, k)$ or $h_N(i, j) < h_N(j, k)$ then the triplet $ij|k$ is consistent with $N$ [7]. The above two items imply that the following Integer Programming (IP) $IP(\tau, s)$ is established for a given triplets $\tau$ with $|L(\tau)| = n$ [7].

$$
\begin{aligned}
\textit{Maximize} \quad & \Sigma_{1 \leq i, j \leq n} h(i, j) \\
\text{Subject to} \quad & h(i, k) - h(i, j) > 0 \quad ij|k \in \tau \\
& h(i, k) - h(i, j) > 0 \quad ij|k \in \tau \\
& 0 < h(i, j) \leq s \quad\quad\ 1 \leq i, j \leq n.
\end{aligned}
$$

The solution of the above IP provides a criterion to obtain the basic tree structure. Ideally it is expected that the above IP has a feasible solution i.e. a solution that satisfies all its constraints. If there is a tree consistent with a given $\tau$ then the above IP has a feasible solution and the solution that maximizes the above IP is the height function of a tree that is consistent with $\tau$. More precisely in this case $h_{T_\tau}$ is the unique optimal solution to the IP $(\tau, l_{G_\tau} + 1)$ in which $T_\tau$ is the unique tree that is constructed by BUILD [7]. If the set of triplets $\tau$ be consistent with a tree, HBUILD can also give the same tree. So in this case by using HBUILD the desired tree consistent with $\tau$ can be constructed in polynomial time based on the optimal solution [7]. Fig 13 indicates an example of the HBUILD process for the given $\tau = \{cd|b, cd|a, bd|a, bc|a\}$.

Generally the above IP has a feasible solution iff the graph $G_\tau$ is a DAG and in this case the minimum $s$ that gives a feasible solution for $IP(\tau, s)$ is $l_{G_\tau} + 1$ [7]. So for a given $\tau$ the IP might have a feasible solution although there is no tree consistent with $\tau$. In the worst case, there is no tree consistent with a given $\tau$ and no feasible solution for the above IP i.e. equivalently the graph $G_\tau$ is not a DAG. To overcome this flaw, the goal is to remove minimum number of edges from $G_\tau$ (minimum number of criterions from the IP) to lose minimum information. The problem of removing minimum number of edges from a directed graph to obtain a DAG is known as the Minimum Feedback Arc Set problem, MFAS problem for short. MFAS is NP-hard [22]. The heuristic method that is introduced in [16] is used to obtain a DAG from $G_\tau$ as follow:

The nodes with in-degree zero cannot participate in any directed cycle. So these nodes are removed and this process is continued in the remaining graph until there is no node with in-degree zero. Similarly, this process is performed for the nodes with out-degree zero in the remaining graph [16].

In the resulting graph that contains no node with in-degree zero or out-degree zero, first color white is assigned to each node. Then for each node $v \in V(G)$ the following is done. Suppose that the out-degree of $v$ is $m$ and $vv_1, vv_2, \ldots, vv_m$ be $m$ such directed edges with $v$ as their tail. These $m$ edges are removed from the resulting graph. The color of $v$ and $v_1, v_2, \ldots, v_m$ are converted to black. For each $v_i$, $1 \leq i \leq m$ the $v_i u$ edge that the color of $u$ is white is removed. Then the color of each node $u$ that is the head of some $v_i$, $1 \leq i \leq m$ is converted to black. The process of removing the edges is continued in a way that the color of all nodes becomes black. The Value of $v$ is defined as the number of remaining edges in the resulting graph. The remaining edges related to the node with the minimum value is removed from $G$ and the resulting graph is a DAG. Fig 14 shows an example of this process [16].

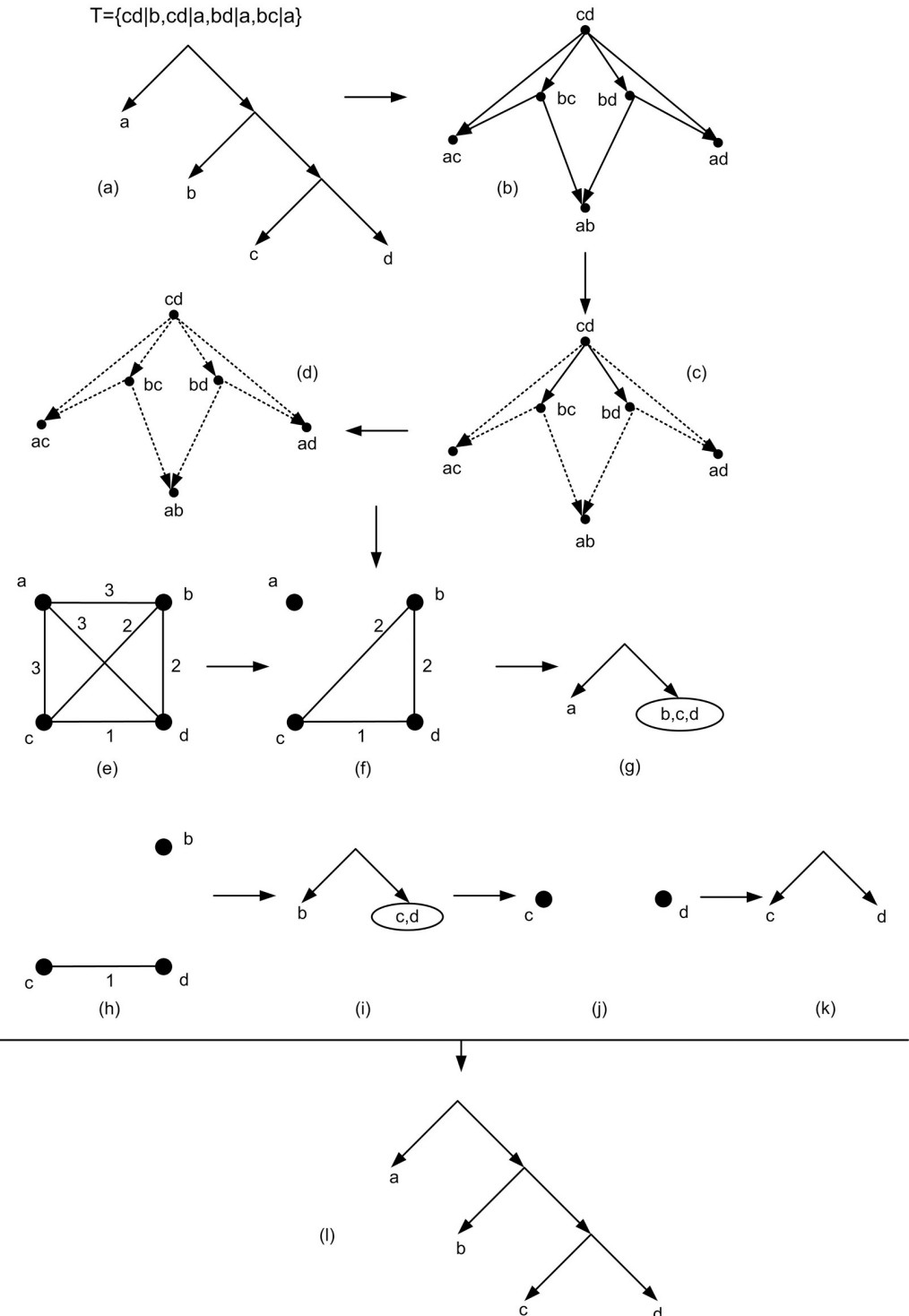

**Fig 13. The HBUILD process for $\tau = \{cd|b, cd|a, bd|a, bc|a\}$.** (a) Firstly a tree $T$ is considered. Then the set $\tau$ of all triplets consistent with $T$ is obtained. (b) $G_\tau$ is obtained from $\tau$. $l_{G_\tau} = 2$ and $h(a, b) = h(a, c) = h(a, d) = 3$ since the out-degree of the nodes $ab$, $ac$, $ad$ are zero. (c) The nodes with out-degree zero and their related edges are removed (removed edges are dashed). $h(b, c) = h(b, d) = 2$. (d) The nodes with out-degree zero and their related edges are removed. $h(c, d) = 1$. (e) The weighted complete graph $(G, h)$ related to the obtained height function in which the edge $\{i, j\}$ has weight $h(i, j)$. (f) Removing the edges with maximum weight from $(G, h)$. (g) The tree that is obtained based on the graph of Fig 13f. (h) Removing the edges with maximum weight from the resulting graph on the set of nodes $b, c, d$. (i) The tree that is obtained based on the graph of Fig 13h. (j) Removing the edges with maximum weight from the resulting graph on the set of nodes $c, d$. (k) The tree that is obtained based on the graph of Fig 13j. (l) The final tree consistent with given $\tau$ that is obtained from HBUILD algorithm by reversing its steps.

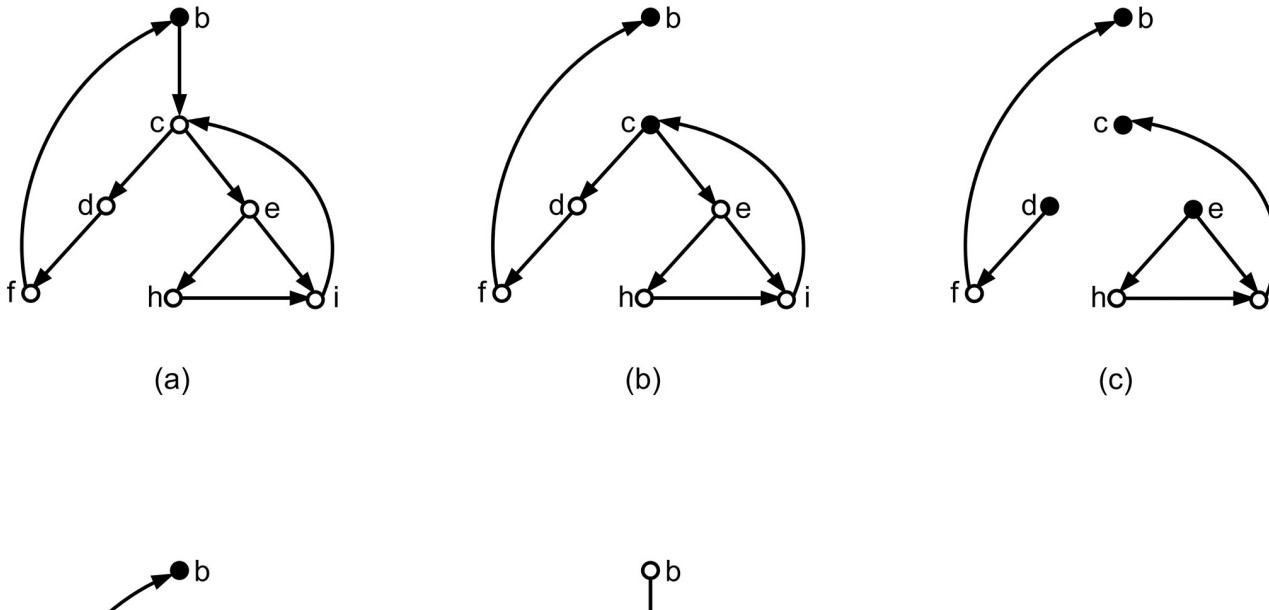

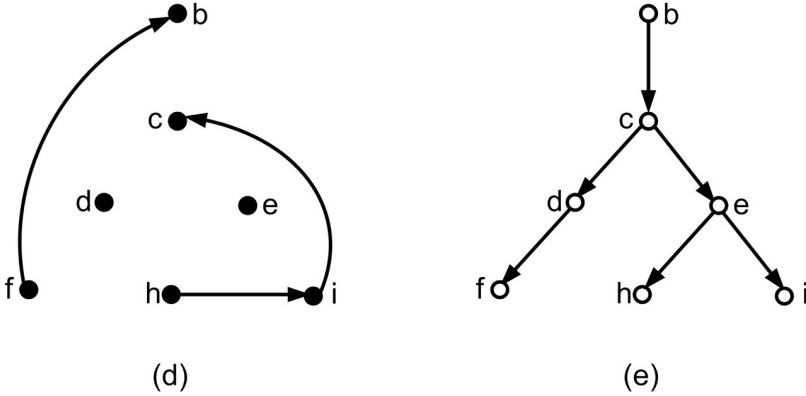

**Fig 14. The process of assigning black and white color to the nodes of a graph with no node with in-degree zero or out-degree zero.** (a) The first step of nodes coloring, by considering the node b as the starting point. (b-d) The next three steps of the nodes coloring. (e) A DAG is obtained from Fig 14a by removing the edges that are determined in the graph of Fig 14d.

For simplicity the new graph that is a DAG is called $G_\tau$ again. Now the height function $h_{G_\tau}$ related to $G_\tau$ is the desired solution.

## Obtaining tree

In the following the goal is to obtain a tree structure from the obtained $h_{G_\tau}$. In the initial step HBUILD is applied on $h_{G_\tau}$. The ideal situation is when HBUILD continues until a tree structure is obtained. However, HBUILD may stop in one of its subsequence steps. More precisely, Let $(G, h)$ be the weighted complete graph related to $h_{G_\tau}$. HBUILD algorithm removes the edges with maximum weight from $(G, h)$. If by removing the edges with maximum weight from each connected component, the resulting graph becomes disconnected then this process continues iteratively until each connected component contains only one node. The basic tree structure is obtained by reversing the above disconnecting process in HBUILD (See Fig 13).

If by removing the edges with maximum weight from a connected component $C$, the resulting graph $C'$ remains connected, then HBUILD halts. Hence, the goal is to disconnect the

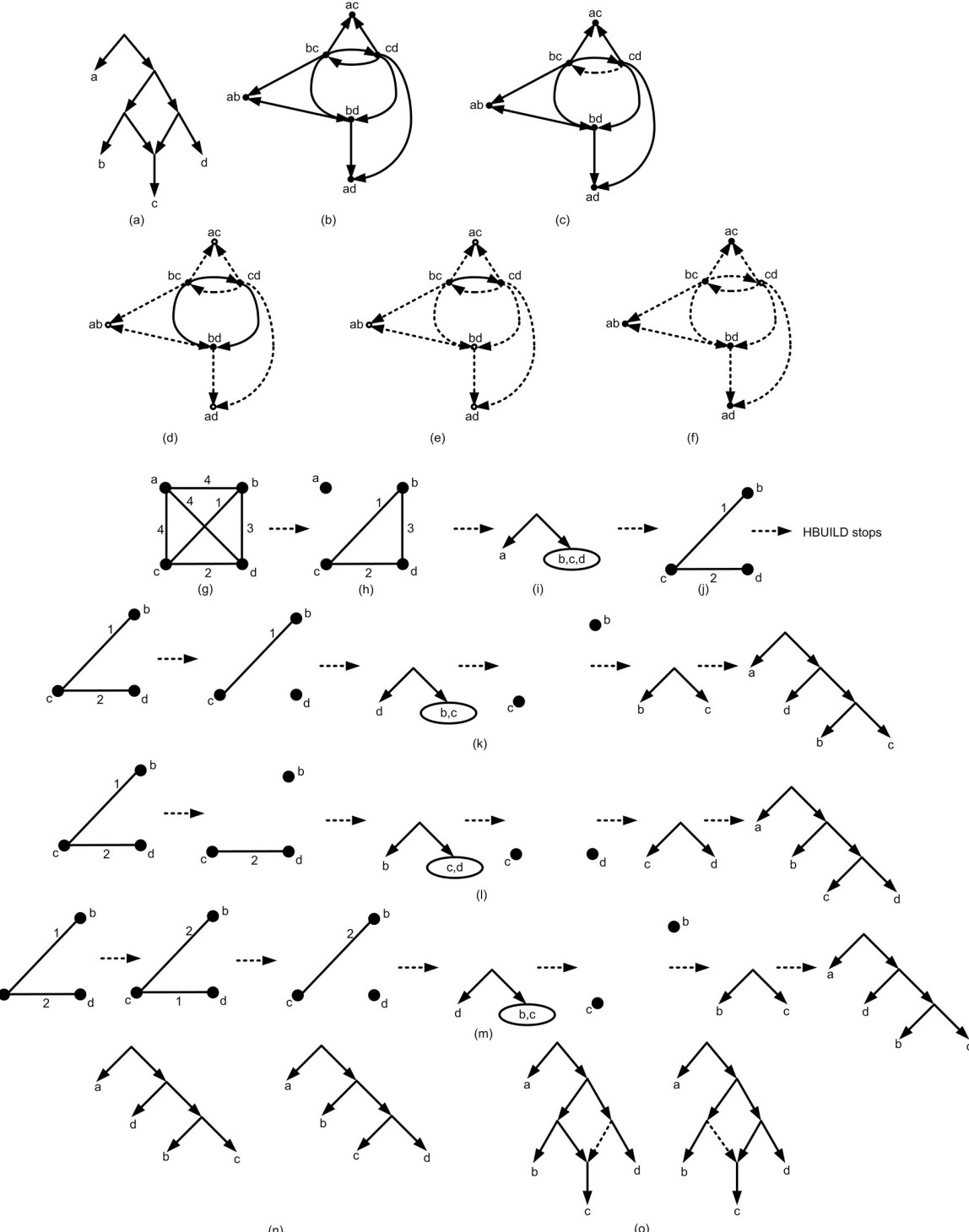

**Fig 15. A basic tree structure construction process for a given** $\tau$ = {**bc|a, bd|a, cd|a, bc|d, cd|b**}. $\tau$ is not consistent with a tree and HBUILD stops in some steps. (a) $\tau$ is obtained from the given network and the network is the optimal network consistent with $\tau$. (b) $G_\tau$ related to $\tau$. (c) $G_\tau$ is not a DAG and contains a cycle. The edge (cd, bc) is removed to obtain a DAG. For simplicity the new DAG is called $G_\tau$ again. $l_{G_\tau} = 3$. (d) The nodes ab, ac, ad with out-degree zero and their related edges are removed. $h(a, b) = h(a, c) = h(a, d) = 4$. (e) The node bd with out-degree zero and its related edges are removed. $h(b, d) = 3$. (f) The node cd with out-degree zero and its related edges are removed. $h(c, d) = 2$. Finally the node bc with out-degree zero is removed. $h(b, c) = 1$. (g) The graph $(G, h)$ based on the obtained $h_{G_\tau}$ related to $G_\tau$. (h) The edges with maximum weight are removed from $(G, h)$. The resulting graph is disconnected. So HBUILD continues.

(i) The tree structure related to the Fig 15h. (j) The edges with maximum weight are removed from the graph of Fig 15h. The resulting graph is still connected. So HBUILD stops. From here three criterions are applied to disconnect the connected subgraphs. (k) To disconnect the connected component of Fig 15j the process of removing the edges with maximum weight (the edges with weight 2) continues. The resulting graph becomes disconnected. In the remaining steps, HBUILD is applied and finally a tree structure is obtained by reversing the steps of the algorithm. (l) To disconnect the connected component of Fig 15j, Min-Cut is applied. So the graph nodes are partitioned into two parts {c, d} and {b} and the edge cb is removed. The resulting graph becomes disconnected. In the remaining steps, HBUILD is applied and finally a tree structure is obtained by reversing the steps of the algorithm. (m) To disconnect the connected component of Fig 15j, Max-Cut is applied. So the graph nodes are partitioned into two parts {c, b} and {d} and the edge cd is removed. The resulting graph becomes disconnected. In the remaining steps, HBUILD is applied and finally a tree structure is obtained by reversing the steps of the algorithm. (n) The two tree structures that finally are obtained (Three tree structures are obtained. the two structures are the same). (o) The two obtained tree structures of Fig 15n are two spanning tree structures of Fig 15a and the algorithm finally obtained these two tree structures.

obtained connected component ($C'$). In order to disconnect $C'$, similar to RPNCH [8] three different processes can be performed as follow (See Fig 15):

I.  The process of removing the edges with maximum weights from $C'$ is continued until $C'$ becomes disconnected.

II. The Min-Cut method is applied on $C'$. Min-Cut is a method that removes minimum weights sum of removed edges in a way that the resulting graph is converted into two connected components [23].

III. Let $w$ be maximum weight of all edges in $C'$. The new weights are computed based on the current weights and $w$. For each edge with weight $m$, its new weight is assigned as:

$$m_{new} = w - m + 1.$$

Then Min-Cut method is applied on the updated graph.

In this research, for each connected component, the above three processes is applied and then by using HBUILD, three possible tree structures are obtained. From here, without loss of generality, the symbol $T_{int}$ is used to show the tree structure obtained from HBUILD or the three tree structures gained from the above processes.

## Binarization

Let $T$ be a rooted tree and $\tau(T)$ be the set of triplets consistent with $T$. Also let $T_{binary}$ be a binarization of $T$ and $\tau(T_{binary})$ be the set of triplets consistent with $T_{binary}$. Then $\tau(T) \subseteq \tau(T_{binary})$. It means that binarization is an effective tool to make the tree structure more consistent with the given triplets. To perform binarization on $T_{int}$, the following heuristic algorithm is proposed.

For a given set of triplets $\tau$ and $T_{int}$ a binary tree structure $T_{intBin}$ is demanded. Binarization can be performed simply with a random approach [7, 8]. In order to make binarization more efficient, a new heuristic algorithm is introduced innovatively in this research. This algorithm is originally based on the three parameters, $w$, $t$, and $p$ [16, 24].

Let $\tau$ be a set of triplets and $V_i, V_j \subseteq L(\tau)$ and $V_i \cap V_j = \emptyset$. Let $W(V_i, V_j) = \{v_i v_j | v \in \tau \mid v_i \in V_i, v_j \in V_j \text{ and } v \notin V_i \cup V_j\}$, $P(V_i, V_j) = \{v_i v | v_j \in \tau \text{ or } v_j v | v_i \in \tau \mid v_i \in V_i, v_j \in V_j \text{ and } v \notin V_i \cup V_j\}$, and $T(V_i, V_j) = \{v_i v_j | v \in \tau \mid v_i \in V_i, v_j \in V_j\}$. Also let $w(V_i, V_j)$, $p(V_i, V_j)$, and $t(V_i, V_j)$ be the cardinality of $W(V_i, V_j)$, $P(V_i, V_j)$, and $T(V_i, V_j)$, respectively [16, 24]. Based on the three parameters ($w$, $t$, and $p$), nine different measure are defined [5]. The measures $M = \{m_1, m_2, \ldots, m_9\}$ are defined as: $m_1 = t(V_i, V_j)$, $m_2 = w(V_i, V_j)$, $m_3 = (w - p)(V_i, V_j)$, $m_4 = \frac{w}{w+p}(V_i, V_j)$,

$m_5 = \frac{w}{t}(V_i, V_j), m_6 = \frac{w-p}{w+p}(V_i, V_j), m_7 = \frac{w-p}{t}(V_i, V_j), m_8 = \left(w - p + \frac{w}{t}\right)(V_i, V_j),$

$m_9 = \left(\frac{w-p}{t} + \frac{w}{w+p}\right)(V_i, V_j).$ By using these measures, nine binary tree structures ($T_{intBin}$) are built from $T_{int}$.

The binarization process is performed as follow:

Binarization Pseudocode

```
 1:  Input: T_int
 2:    T_intBin = T_int
 3:    If T_intBin is binary
 4:      Do nothing
 5:    else
 6:      for each vertex v from T_intBin with c_1, c_2, ..., c_n children, n > 2.
 7:        Initialize a set C with {c_1, c_2, ..., c_n}.
 8:        while |C| > 1 do
 9:          Find and remove two vertices c_i, c_j ∈ C with maximum measure
             values.
10:            Merge c_i and c_j and obtain c_new0.
11:            Generate 6 new structures using SPR with roots c_new1, c_new2,
             ..., c_new6.
12:            Among 6+1 structures, select the more consistent struc-
             ture and add its root to C.
13:            Update T_intBin respect to selected structure.
14:          end while
15:        end for
16:      Output: T_intBin
```

The binarization process is performed based on using nine different defined measures and Subtree- Pruning-Regrafting (SPR) [25, 26]. SPR is a method in tree topology search [25]. In the binarization process SPR helps to obtain a tree from $T_{int}$ more consistent with input triplets. If $T_{int}$ is binary there is nothing to do; Else there is at least a vertex $v$ in $T_{int}$ with $c_1, c_2, \ldots, c_n$ children, $n > 2$. In this case the goal is replacing this part of the tree with a binary structure (a binary subtree). For this purpose in the first step there are $n$ sets each contains one $c_i$, $1 \leq i \leq n$. Then iteratively in each step, two sets with the maximum measure values (according to the one of the nine defined measures) are selected. Let $c_i$ and $c_j$, $1 \leq i, j \leq n$ be two nodes with maximum measure value. By merging $c_i$ and $c_j$, a new vertex $c_{new}$ is created (See Fig 16). Here, SPR is used innovatively to improve the merging consistency.

Suppose that $c_{lk}$ and $c_{rk}$ are the roots of the two left and right subtrees of $c_k$, $k \in \{i, j\}$. The idea behind SPR is replacing subtrees to achieve a new binary tree structure with higher consistency. In this work the potential replacement are introduced in six different ways as follow (See Fig 17). *i*) $c_i \rightleftharpoons c_{lj}$, *ii*) $c_i \rightleftharpoons c_{rj}$, *iii*) $c_j \rightleftharpoons c_{li}$, *iv*) $c_j \rightleftharpoons c_{ri}$, *v*) $c_{li} \rightleftharpoons c_{lj}$, *vi*) $c_{li} \rightleftharpoons c_{rj}$. By using these SPRs, six new structures are obtained. Among these tree structures and the structure without replacement, the best tree structure consistent with more input triplets is selected.

## Network construction

Let $\tau' \subseteq \tau$ be the set of triplets that are not consistent with $T_{intBin}$. Here, the goal is to add some edges to $T_{intBin}$ in order to construct a network consistent with input $\tau'$. In the network construction process, edges are added incrementally to obtain the final network consistent with $\tau$. In order to add edges, we use innovatively a heuristic criterion to select edges rather than random selection. The heuristic criterion is depended on the current non-consistent triplets $\in \tau'$ and the current network structure. To this purpose, for each pair of edges of the current network structure, a value is assigned. To compute the value of each pair $\{e, f\}$ of edges, a new edge is added by connecting $e$ and $f$ via two new nodes $n_e$, $n_f$ (See Fig 18). The value is the

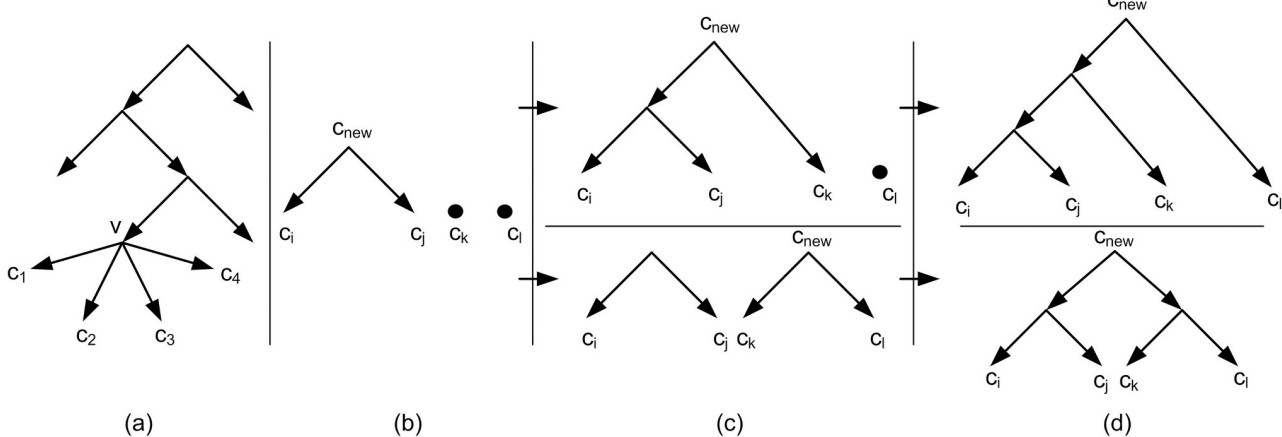

**Fig 16.** (a) A non-binary tree. The node $v$ contains four children $c_1, c_2, c_3, c_4$. (b) The nodes $c_i, c_j, c_k, c_l$ are an arbitrary relabeling of the nodes $c_1, c_2, c_3, c_4$. Firstly two nodes are merged. (c) Secondly two structures can be obtained from the structure of Fig 16b. (d) finally for each structure of Fig 16c, a binary structure is obtained. These two binary structures are replaced with the non-binary part of the tree of Fig 16a.

number of triplets in $\tau'$ that are consistent with the new network structure. In each step of adding an edge, the set of triples $\tau'$, are updated by removing consistent triplets.

## Time complexity

In this section, we investigate the time complexity of Netcombin. For the input triplets $\tau$ let $|L(\tau)| = n$ and $|\tau| = m$. At first $G_\tau$ should be computed. Its time complexity is $O(m)$. Then, if $G_\tau$ is not a DAG the heuristic algorithm is applied to make it a DAG. For each node, the computation of Value is performed in $O(m)$. Therefore for all nodes it needs $m \times O(m) = O(m^2)$.

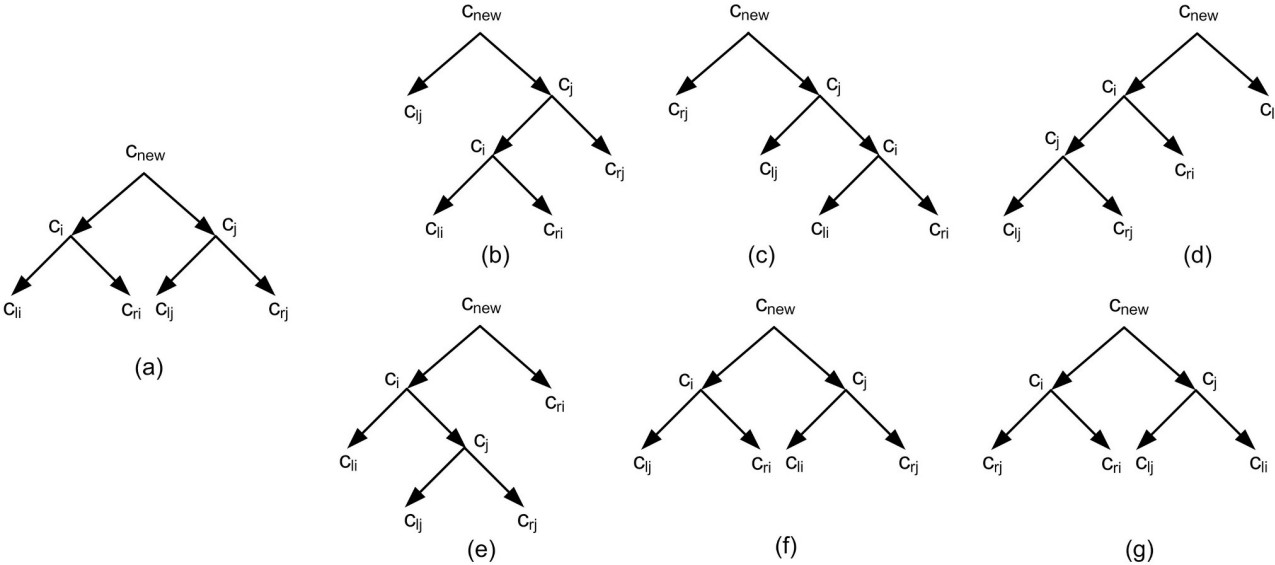

**Fig 17. SPR is used to obtain six different structures from a given tree structure.** (a) The structure that is obtained by merging $c_i, c_j$ and connecting them to a new node $c_{new}$. (b to g) Six different tree structures that are obtained from Fig 17a and by using SPR with replacing $c_i \rightleftharpoons c_{lj}, c_i \rightleftharpoons c_{rj}, c_j \rightleftharpoons c_{li}, c_j \rightleftharpoons c_{ri}, c_{li} \rightleftharpoons c_{lj}, c_{li} \rightleftharpoons c_{rj}$, respectively.

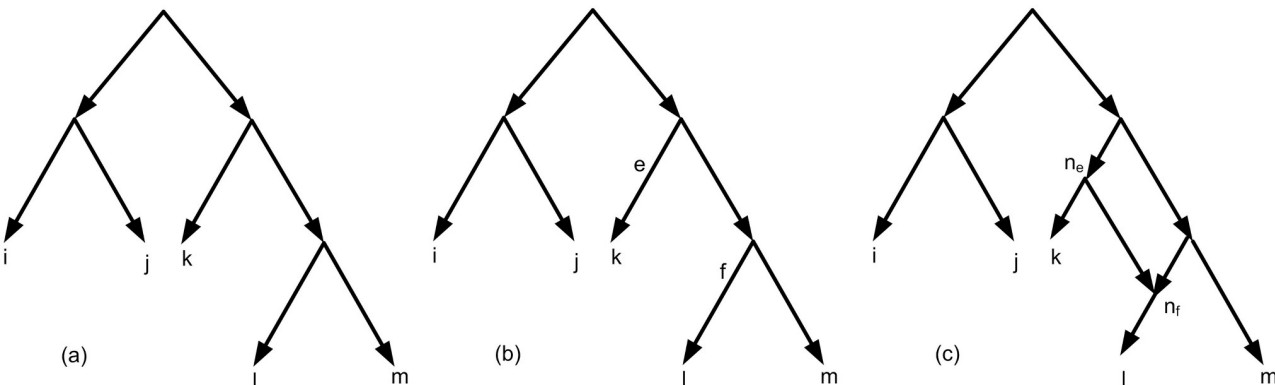

**Fig 18.** (a) $T_{intBin}$ for $\tau' = \{ij|k,\, ij|l,\, ij|m,\, lm|k,\, lm|j,\, lm|i,\, mk|i,\, mk|j,\, lk|i,\, lk|j\}$ and $\tau = \tau' \cup \{lk|m\}$. (b) Two edges $e$ and $f$ are selected to obtain a network consistent with $\tau$. (c) Final network consistent with $\tau$.

To obtain a tree, in method I, each step of removing the edges with maximum weight is done for each connected component in $O(m)$. Also in each step the number of connected components should be compared with previous step. Thus Depth First Search (DFS) algorithm is performed in $O(n)$. The overall runtime is $O(mn)$. Since there are $n$ nodes, the total runtime is $O(mn^2)$.

For the method II, in each step, it takes $O(mn)$ to remove the edges with maximum weight. Then Min-Cut is performed in $O(mn + n^2 \, logn)$. The overall run time is $O(mn + n^2 \, logn)$. There are $n$ nodes and the total runtime is $O(mn^2 + n^3 \, logn)$.

The runtime of the method III is the same as the method II. So obtaining tree $T_{int}$ is performed in $O(mn^2 + n^3 \, logn) + O(m^2)$.

The binarization process to obtain each $T_{intBin}$ is computed in $O(mn^3)$ [4, 24]. The time complexity for obtaining all 27 binary trees $T_{intBin}$ is $27 \times O(mn^3)$ which is equal to $O(mn^3)$.

Finally the network construction runtime is as follow: The number of edges of $T_{intBin}$ is $O(n)$. Also at most $O(m)$ edges are added to obtain the final network. So the number of edges of the final network is $O(m + n)$. Investigating the consistency of the new network (with the remaining triplets) which is obtained from the previous network and by adding a new edge is done in $O((m + n)^2)$. Since there are $O(m + n)$ edges so in each step the runtime of adding a new edge is $O((m + n)^3)$. This process is done at most $m$ times. So the total runtime of this step is $O(m(m + n)^3)$.

Finally the Netcombin runtime is $O(mn^2 + n^3 \, logn) + O(m^2) + O(mn^3) + O(m(m + n)^3) \in O(m(m + n)^3 + n^3 \, logn)$.

## Experiments

The RPNCH, NCHB, SIMPLISTIC and TripNet are famous algorithms in constructing phylogenetic networks from given triplets. The SIMPLISTIC algorithm just works for dense triplets [6], while there is no constraints on the NCHB, TripNet, and RPNCH inputs [7, 8, 16]. In order to evaluate the performance of Netcombin, the following scenario is designed.

### Data generation

There is two standard approaches to generate triplets data. Firstly, triplets can be generated randomly which is the simplest way. Secondly, triplets can be obtained from sequences data. Sequences data usually are in the form of biological sequences. Biological sequences can be

obtained from species or from simulation software that can generate these kinds of sequences under biological assumptions. In this research we used the second approach using a simulation software. There are standard methods for converting sequences into triplets. Maximum Likelihood (ML) is the well-known method which constructs tree from sequence data [5, 6]. For this reason, TREEVOLVE is used which is a software for generating biological sequences [27]. TREEVOLVE has different parameters that can be adjusted manually. In this research we set the parameters, the *number of samples*, the *number of sequences*, and the *length of sequences*. For the other parameters, the default values are used. The *number of sequences* (number of leaf labels) is set to 10, 20 30, and 40 and the *length of sequences* is set to 100, 200, 300, and 400. For each case, the *number of samples* is 10. So totally 160 different sets of sequences are generated. Then PhyML software is used which works based on Maximum Likelihood (ML) criterion. For each set of sequences, all subsets of three sequences are considered and for each of them, an outgroup is assigned. Each subset of three sequences plus the assigned outgroup, are considered as input for PhyML and for these data the output of PhyML is a quartet. Finally by removing the outgroup from each quartet, the set of triplets is obtained. In this research, each triplet information related to a quartet in which the weight of its unique inner edges is zero, is removed. This is because of these types of triplets contains no information and are stars. The way of generating triplets may give non-dense sets of triplets. SIMPLISTIC is used as a method for comparison and its output should be dense. So by adding a random triplet correspond to each star, each non-dense set is converted to a dense set and is used as the input.

## Experimental results

In order to show the performance of Netcombin we compare it with TripNet, SIMPLISTIC, NCHB, and RPNCH on the data that are generated in the previous subsection. Since for large size data, SIMPLISTIC has not the ability to return a network in an appropriate time, the time restriction 6 hours is considered. Let $N_{finite}$ be the set of networks for which the running time of the method is at most 6 hours. Let $S_{sequence}$ shows the number of sequences where $S_{sequence} \in$ {10, 20, 30, 40}. The output of TripNet, SIMPLISTIC, NCHB, and RPNCH is a unique network, but Netcombin outputs 27 networks and the best network is reported. Since the process of constructing these 27 Netcombin networks is independent, we apply Netcombin in a parallel way to obtain 27 networks simultaneously. In implementation we used a PC with Corei7 CPU and run our algorithm on its cores in parallel.

The results of comparing these methods on the two optimality criterions and running time are available in Tables 1 to 4.

Table 1 and 2 show the results of the number of networks that belong to $N_{finite}$, and the average of running time of the networks that belong to $N_{finite}$. These results show that when the number of taxa is 10, in all cases, all methods on average give an output in at most 2 seconds. When the number of taxa is 20, in 5% of the cases, SIMPLISTIC has not the ability to return a network in less than 6 hours. For the remaining 95% of the cases SIMPLISTIC on

**Table 1. The number of Netcombin, TripNet, NCHB, SIMPLISTIC, and RPNCH networks that belong to $N_{finite}$.**

| Number of sequences ($s_{sequence}$) | 10 | 20 | 30 | 40 |
|---|---|---|---|---|
| Number of the Netcombin networks $\in N_{finite}$ | 40 | 40 | 40 | 40 |
| Number of the TripNet networks $\in N_{finite}$ | 40 | 40 | 40 | 40 |
| Number of the NCHB networks $\in N_{finite}$ | 40 | 40 | 40 | 40 |
| Number of the SIMPLISTIC networks $\in N_{finite}$ | 40 | 38 | 13 | 0 |
| Number of the RPNCH networks $\in N_{finite}$ | 40 | 40 | 40 | 40 |

**Table 2. The average running time results for the networks that belong to $N_{finite}$ for Netcombin, TripNet, NCHB, SIMPLISTIC, and RPNCH.**

| Number of sequences ($s_{sequence}$) | 10 | 20 | 30 | 40 |
|---|---|---|---|---|
| Netcombin avg running time for networks $\in N_{finite}$ (Sec) | 2 | 4 | 15 | 44 |
| TripNet avg running time for networks $\in N_{finite}$ (Sec) | 1 | 1.7 | 210 | 740 |
| NCHB avg running time for networks $\in N_{finite}$ (Sec) | 1 | 1.8 | 203 | 745 |
| SIMPLISTIC avg running time for networks $\in N_{finite}$ (Sec) | 1 | 310 | 2600 | - |
| RPNCH avg running time for networks $\in N_{finite}$ (Sec) | 1 | 2 | 10 | 30 |

**Table 3. The average number of reticulation nodes (rets for short in table) results for the networks that belong to $N_{finite}$ for Netcombin, TripNet, NCHB, SIMPLISTIC, and RPNCH.**

| Number of sequences ($s_{sequence}$) | 10 | 20 | 30 | 40 |
|---|---|---|---|---|
| Netcombin avg number of rets for networks $\in N_{finite}$ | 2 | 4 | 9 | 15.5 |
| TripNet avg number of rets for networks $\in N_{finite}$ | 0.9 | 2.6 | 8 | 16.3 |
| NCHB avg number of rets for networks $\in N_{finite}$ | 0.7 | 1.8 | 7.2 | 15.2 |
| SIMPLISTIC avg number of rets for networks $\in N_{finite}$ | 2.325 | 6.95 | 11.275 | - |
| RPNCH avg number of rets for networks $\in N_{finite}$ | 3 | 9 | 13 | 20 |

average gives an output in 310 seconds. For these data the other four methods on average construct a network in less than 4 seconds. When the number of taxa is 30, in 32.5% of the cases, on average SIMPLISTIC outputs a network in 2600 seconds. For the remaining 77.5% of the cases, SIMPLISTIC has not the ability to return a network in less than 6 hours. For these data, on average Netcombin and RPNCH output a network in at most 15 seconds, while NCHB and TripNet on average output a network in 203 and 210 seconds, respectively. When the number of input taxa is 40, in all cases SIMPLISTIC does not return an output in time restriction 6 hours. In this case Netcombin and RPNCH on average output a network in at most 44 seconds while NCHB and TripNet return a network in time at least 740 seconds.

Tables 3 and 4 indicate the results for the two optimality criterions i.e. the number of reticulation nodes and level for the networks that belong to $N_{finite}$. The results show that when the number of taxa is 10, on average the number of reticulation nodes for the TripNet and NCHB networks is at most 0.9, while for these data on average the Netcombin, RPNCH, and SIMPLISTIC number of reticulation nodes is at least 2 and at most 3. Also for these data, on average the level of the NCHB and TripNet networks, is not more than 0.9, while the level of Netcombin, SIMPLISTIC, and RPNCH networks, on average is at least 2 and at most 2.8. When the number of input taxa is 20 on average the TripNet and NCHB number of reticulation nodes is 2.6 and 1.8, respectively. For these data the Netcombin number of reticulation nodes on average is 4,

**Table 4. The average level results for the networks that belong to $N_{finite}$ for Netcombin, TripNet, NCHB, SIMPLISTIC, and RPNCH.**

| Number of sequences ($s_{sequence}$) | 10 | 20 | 30 | 40 |
|---|---|---|---|---|
| Netcombin avg level for networks $\in N_{finite}$ | 2 | 3 | 7.4 | 15 |
| TripNet avg level for networks $\in N_{finite}$ | 0.9 | 2.3 | 6.9 | 16 |
| NCHB avg level for networks $\in N_{finite}$ | 0.7 | 1.5 | 6 | 15 |
| SIMPLISTIC avg level for networks $\in N_{finite}$ | 2.05 | 4.2 | 6.95 | - |
| RPNCH avg level for networks $\in N_{finite}$ | 2.8 | 6.4 | 10.5 | 19 |

while for SIMPLISTIC and RPNCH, on average the number of reticulation nodes is 6.95 and 9, respectively. Also for these data the level of the NCHB and TripNet networks on average are 1.5 and 2.3, respectively. For these data on average the level of the Netcombin networks is 3, while for the SIMPLISTIC and RPNCH networks the level is 4.2 and 6.4, respectively. When the number of taxa is 30, on average the number of NCHB, TripNet, and Netcombin reticulation nodes, are at least 7.2 and at most 9, while for the SIMPLISTIC and RPNCH networks on average this number is 11.275 and 13, respectively. For these data on average the level of the NCHB, TripNet, Netcombin, and SIMPLISTIC networks is at least 6 and at most 7.4 while on average the RPNCH networks level is 10.5. When the number of taxa is 40, on average the, NCHB, Netcombin, and TripNet number of reticulation nodes are 15.2, 15.5, and 16.3, respectively, while RPNCH networks on average contain 20 reticulation nodes. For these data on average the level of Netcombin, NCHB, and TripNet netowrks are 15, 15, and 16, respectively, while the level of the RPNCH networks on average is 19.

## Discussion

In this paper we investigated the problem of constructing an optimal network consistent with a given set of triplets. Minimizing the level or minimizing the number of reticulation nodes are the two optimality criterion. This problem is known to be NP-hard [17, 18]. By analyzing existing research we can divide the solution of constructing networks based on triplets, into two approaches. In the first approach, the reticulation nodes are recognized and then are removed from the set of taxa and a tree structure is obtained for the remaining taxa. Finally the network consistent with all given triplets is obtained by adding reticulation nodes to the tree structure. In the second approach, a tree structure is obtained and then by adding new edges to the tree structure, the final network consistent with all triplets is obtained. SIMPLISTIC [6], TripNet [7] and NCHB [16] belong to the first approach and RPNCH [8] belongs to the second approach. According to our best of knowledge, all the researches on this problem fall into one of these approaches. Therefore, in recent papers researchers try to improve these approaches gradually. It means that each improvement is valuable because it can reduce the time and costs, effectively. In this paper we introduced Netcombin which is a method for producing an optimal network consistent with a given set of triplets. In order to show the performance of Netcombin we compared it with NCHB, TripNet, SIMPLISTIC, and RPNCH on the 160 different sets of triplets that are generated in the process that is introduced in subsection 4-1.

The results show that although, on average RPNCH is the fastest method, but the level and the number of reticulation nodes of its results are highest. More over on average the differences between Netcombin, NCHB, and TripNet results for the two optimality criterions with RPNCH results are significant.

The results show that on average for small size data SIMPLISTIC is appropriate. But by increasing the number of taxa and for large size data it has not the ability to return a network in an appropriate time and its running time is highest. Also in all cases on average the SIMPLISTIC number of reticulation nodes and levels are just better than RPNCH. Note that SIMPLISTIC just works for dense sets of input triplets. The results show that by increasing the number of taxa, the running time of SIMPLISTIC increases exponentially. In more details when the number of taxa is 40, in time less than 6 hours it does not return any network, while the other 4 methods in at most 745 seconds output a network.

Also the results show that on average NCHB and TripNet running time results are nearly the same, but on average the two optimality criterions for NCHB results are better compared to TripNet. Note that the differences between TripNet and NCHB results for the optimality criterions are not significant.

The results show that for small size data TripNet and NCHB are appropriate and their results for the optimality criterions and running time are on average the best. But by increasing the number of taxa, the running time of these methods exceeds significantly compared to Netcombin, while the two optimality criterions for their networks are nearly the same with Netcombin networks results.

The results show that generally and by considering the running time, the level, and the number of reticulation nodes of the final networks, on average Netcombin is a valuable method that returns reasonable network in an appropriate time.

## Supporting information

**S1 File.**
(ZIP)

**S2 File.**
(ZIP)

**S3 File.**
(ZIP)

**S4 File.**
(ZIP)

## Acknowledgments

The first author would like to thank the Institute for Research in Fundamental Sciences (IPM), Tehran, Iran. "*The authors declare no conflict of interest.*"

## Author Contributions

**Data curation:** Hadi Poormohammadi, Mohsen Sardari Zarchi.

**Formal analysis:** Hadi Poormohammadi, Mohsen Sardari Zarchi.

**Investigation:** Hadi Poormohammadi.

**Methodology:** Hadi Poormohammadi.

**Project administration:** Hadi Poormohammadi.

**Software:** Mohsen Sardari Zarchi.

**Supervision:** Mohsen Sardari Zarchi.

**Writing – original draft:** Hadi Poormohammadi, Mohsen Sardari Zarchi.

**Writing – review & editing:** Hadi Poormohammadi, Mohsen Sardari Zarchi.

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
