## [Decision Letter · Decision Letter 0]

12 Feb 2020

PONE-D-19-34247

NetCombin‎: ‎An algorithm for optimal level-k network construction from triplets

PLOS ONE

Dear Dr. Poormohammadi,

Thank you for submitting your manuscript to PLOS ONE. After careful consideration, we feel that it has merit but does not fully meet PLOS ONE’s publication criteria as it currently stands. Therefore, we invite you to submit a revised version of the manuscript that addresses the points raised during the review process.

We would appreciate receiving your revised manuscript by Mar 28 2020 11:59PM. To enhance the reproducibility of your results, we recommend that if applicable you deposit your laboratory protocols in protocols.io, where a protocol can be assigned its own identifier (DOI) such that it can be cited independently in the future. For instructions see: http://journals.plos.org/plosone/s/submission-guidelines#loc-laboratory-protocols

We look forward to receiving your revised manuscript.

Kind regards,

Hocine Cherifi

Academic Editor

PLOS ONE

Journal Requirements:

"No, the funders had no role in study design, data collection and analysis, decision to publish, or preparation of the manuscript."

Reviewers' comments:

Reviewer's Responses to Questions

**Comments to the Author**

1. Is the manuscript technically sound, and do the data support the conclusions?

Reviewer #1: No

Reviewer #2: Partly

2. Has the statistical analysis been performed appropriately and rigorously? 

Reviewer #1: N/A

Reviewer #2: No

3. Have the authors made all data underlying the findings in their manuscript fully available?

Reviewer #1: No

Reviewer #2: No

4. Is the manuscript presented in an intelligible fashion and written in standard English?

Reviewer #1: No

Reviewer #2: Yes

5. Review Comments to the Author

Reviewer #1: A heuristic method for constructing a phylogenetic network from a set of

rooted triplets is presented.

The problem objective is to output a simplest possible phylogenetic network

that contains all of the specified rooted triplets.

Here, simplest means having the fewest reticulation nodes or the smallest

level.

There are three serious issues with this paper:

1. lack of originality,

2. technical weakness,

3. poor presentation.

For this reason, my recommendation is reject.

More detailed comments are given below.

Issue 1: Lack of originality.

The presented heuristic is very similar to other heuristics previously

published by the authors that first build a phylogenetic tree and then

greedily insert edges into it (thereby turning it into a phylogenetic

network) until all triplets are satisfied.

As far as I can see, there are only two new contributions in this paper.

The first one is to apply Wu's technique to select a good binary phylogenetic

tree to use as the base, instead of just refining non-binary nodes to make

make them binary in some arbitrary way.

The second one is to consider every pair of existing edges when deciding the

insertion point of each new edge in the last step, instead of inserting these

new edges randomly.

Both of these contributions are rather trivial adjustments.

It seems like the authors are trying to publish a new paper every time they

come up with some minor improvement to their own algorithm.

Issue 2: Technical weakness.

The resulting method ("NetCombin") is not analyzed formally.

This is no problem, though, since the authors evaluated its performance by

comparing it experimentally against some other methods on simulated data.

However, these comparisons do not seem fair because in the experiments, they

applied NetCombin in parallel to construct 27 different networks and then

output the best one found.

For each of the other methods, they only constructed one network and output

it.

Thus, the running time of 44 seconds for NetCombin should be reported as

44*27 = 1188 seconds, etc., which means that it's actually slower than the

NCHB method while outputting solutions that are worse than NCHB's.

Furthermore, the experiments only go up to 40 sequences so it's difficult to

draw any conclusions regarding the relative performance of NetCombin,

TripNet, NCHB, and RPNCH.

Issue 3: Poor presentation.

The authors don't even bother to explain all the steps of the method; in many

places, they just refer to some of their old papers.

It would have been better to list the complete algorithm.

Throughout the paper, there is lots of vague phrasing, mistakes, grammatical

errors, and missing references to the literature.

For example:

- The title is very strange.

The parameter k is never defined or used in the paper.

What is k?

Does "optimal level-k network construction" mean that one wants to find a

network that minimizes the value of k?

- Abstract and introduction:

It's better to say "or" instead of "and" when talking about minimizing the

level and minimizing the number of reticulation nodes since they are not the

same thing.

- The abstract claims "The binarization process innovatively uses a measure

to construct a binary rooted tree T consistent with the maximum number of

input triplets.", but this is misleading.

To find such a T is another NP-hard optimization problem and applying Wu's

technique just gives an approximation of "a binary rooted tree consistent

with the maximum number of input triplets", not always an optimal one.

Similar phrasing occurs at the bottom of page 4.

- The abstract proudly declares that T is "expanded in an intellectual

process".

Actually, it's just a straightforward greedy algorithm.

- The abstract says "The experimental results on real data indicate that".

This is not true; only simulated data was used.

- To make the problem definition easier to read, say clearly that all of the

input triplets have to be consistent with the output network and that either

the level or the number of reticulation nodes in the output network is to be

minimized.

- When describing previous related results, it's important to explain how

your old algorithms work since the new algorithm is a modification of them.

My impression is that there is almost no difference, making this paper weak.

- The section "Definitions and notations" starts with the promise "In this

section the basic definitions that are used in the proposed algorithm, are

presented formally.".

A few sentences later, the text begins referring to something called "N",

which has never been introduced.

- The "Method" section says "The height function enforce that the obtained

rooted tree be consistent with maximum number of triplets of tau.".

In case no tree is consistent with all the triplets in tau, the IP won't have

any valid solution, so I don't see how using the height function can yield a

tree that is consistent with the maximum number of triplets in tau.

- How does NetCombin solve the IPs that were set up?

- What is the difference between BUILD and HBUILD?

Line 207 suggests that they are "equivalent", but what does that mean?

- In the experimental results, is the "number of sequences" the same thing as

the "number of leaf labels"?

- Lines 366-404 seem pointless.

The reader can just look at the tables to get that information.

- Does Table 4 list the number of reticulations or the level?

The caption says one thing but the table itself says another thing.

- The example in Figure 5 is wrong.

The input set of triplets contains a triplet de|f, which means that the BUILD

algorithm will join d and e by an edge so that all of the leaf labels end up

in the same connected component.

- The caption of Figure 5 is misleading.

It says "two nodes i,j in X are connected iff" but usually two nodes are said

to be "connected" if there is a path (of any length) between them.

It would be better to say "two nodes i,j in X are adjacent iff" or "two nodes

i,j in X are connected by an edge iff".

- The caption of Figure 5 refers to the "Aho graph".

However, the BUILD algorithm was published in a paper written by four people.

To be fair, call it "the Aho-Sagiv-Szymanski-Ullman graph" or something like

that if you insist on using family names.

- The example in Figure 10 is wrong.

The two trees in (b) and (c) are not binary.

In both trees, the node lca(f,g) has three children.

Also, the text refers to "its two different binarizations" but there are many

more possible binarizations.

- The caption of Figure 14 uses Max-Cut in part (m).

Should it be Min-Cut?

- Lines 321-322 state some strange-looking time complexity (without any

proof) for one of the steps in the algorithm.

It doesn't seem relevant as the time complexity of all the other steps has

been completely ignored.

- More than 30% of the references in the bibliography are self-references.

This is too much.

On the other hand, many historically important bibliographic references are

missing, such as the following.

* Previous work on the problem should also mention:

M. Habib, T.-H. To: "Constructing a minimum phylogenetic network from a dense

triplet set", Journal of Bioinformatics and Computational Biology, 10(5):

1250013, 2012.

* The concept of the level of phylogenetic networks (lines 50-51) comes from:

C. Choy, J. Jansson, K. Sadakane, W.-K. Sung: "Computing the Maximum

Agreement of Phylogenetic Networks", Theoretical Computer Science, 335(1):

93-107, 2005.

* Galled trees were not invented by [15,16] as claimed on line 80 but by:

D. Gusfield, S. Eddhu, C. Langley: "Optimal, efficient reconstruction of

phylogenetic networks with constrained recombination", Journal of

Bioinformatics and Computational Biology, 2(1): 173-213, 2004.

* The Min-Cut method applied to Aho et al.'s BUILD algorithm is from:

L. Gasieniec, J. Jansson, A. Lingas, A. Ostlin: "On the complexity of

constructing evolutionary trees", Journal of Combinatorial Optimization,

3(2-3): 183-197, 1999.

* The SPR technique is much older than reference [21]; see, e.g.:

J. Hein: "Reconstructing evolution of sequences subject to recombination

using parsimony", Mathematical Biosciences, 98: 185-200, 1990.

Reviewer #2: Revision of the paper: NetCombin‎: ‎An algorithm for optimal level-k network construction from triplets

The authors propose a novel phylogenetic tree construction algorithm called NetCombin, introduced to construct an optimal network which is consistent with input triplets. The authors compare their method with state-of-the-art such as RPNCH, NCH and similar and show competitive performance. The paper is overall well written and easy to follow, however, some critical errors need to be corrected in order for it to be publishable (I would consider this a major revision).

1.) The authors e.g., in the paragraph from lines 10-14 discuss the importance of constructing graphs instead of trees. However, the first example are just trees. Further, the terminology is not consistent. Even though each tree is a graph, not each graph is a tree, thus the authors should stick to notation of directed graphs in my opinion. The comment is related to the following text:

“The rooted structures 12 are directed graphs which contains a unique vertex called root with in-degree 0 and 13 out-degree at least 2 .. Figure 1a shows an example of a rooted tree.”

2.) The quality of the second figure is very poor and cut off. Please correct this.

3.) line 48: please define the notion of the level of a network. Even though this is a basic concept, it appears here for the first time without any clarification.

4.) Line 64: So, introducing efficient heuristic methods to solve this problem is demanding.

Perhaps “necessary” instead of “demanding” is meant?

5.) The name of the proposed algorithm is NetCombin, however this is not consistent throughout the paper, please unify.

6.) Line 160. Is N the set of natural numbers with 0 or without?

Comments related to the method:

1.) Line 218, please discuss in more detail on MFAS heuristic used. This is highly relevant to the proposed approach.

2.) Line 230, what are the three “???”? Explain.

3.) Line 282 onwards a couple of lines. The authors provide the pseudocode of the proposed algorithm, however, it is not clear why all 9 different measures for construction of binary trees (as apparently used in [5]). This computational step appears somewhat redundant and seems like everything that can be, is computed. In line 301, authors claim innovative use. Could this be elaborated in more detail? I don’t see how exhaustive enumeration of arbitrary 9 measures is any more innovative than doing this for more measures, unless the 9 offer some form of theoretical grounding that makes this traversal/scoring feasible? Furthermore, how can one resolve possible ties? What if the score distribution is ~uniform? Is the subtree selected at random? Please elaborate on these details.

4.) Line 309 onwards. It seems that the network construction step is rather expensive (n^3, m^2). To what extent can this be made run in parallel. Could you discuss this aspect, at least theoretically?

Results:

I don’t quite understand the reasoning behind: “but NetCombin 360 outputs 27 networks and the best network is reported. Since the process of constructing 361 these 27 NetCombin networks is indepe”

Why is only the best one reported? Isn’t this somewhat non-representative for real-life situations. Could you elaborate on that? Further, do other methods also output multiple networks? How did you select the compared against there?

Further, if you report the average running time, please repreat the experiments at least 5 times and report also the standard deviations.

I am not entirely sure what the measure of success is in T - it shows the number of outputted networks? Is more better? Why?

Further, low runtime is meaningless if the quality of results is low. Comment on the runtime w.r.t T3 and T4.

Discussion:

There is not enough discussion on the results (see previous section). Further, the last line reads as “NetCombin is a 434 valuable method that returns an appropriate network in an appropriate time”. What does that mean? What is appropriate? Do you mean reasonable?

Why would the user prefer NetCombin, if it performs on par with existing state-of-the-art?

Please comment on this issue.

The authors should further discuss why NCHB is a worse alternative, as it seems to perform very similarly. Is the runtime main benefit of NetCombin?

Statistical evaluation:

The authors propose tabelaric comparison of results, however, such results do not offer statistically sound insights into the algorithm's inner workings. I would suggest the authors to consider at least critical distance diagrams if possible.

Availability:

Please, make the code and data simulators (or datasets) publicly available via .e.g, github.

Final remarks:

All images are of terrible quality. Please, render the plots as vector graphics if possible, otherwise use >400 dpi. Further, the authors should specify what exactly are the main adopted novelties at the end of the introductory section.

6. PLOS authors have the option to publish the peer review history of their article (what does this mean?). If published, this will include your full peer review and any attached files.

Reviewer #1: No

Reviewer #2: No

---

## [Author Response · Author response to Decision Letter 0]

20 Aug 2020

Reviewer 1:

"The presented heuristic is very similar to other heuristics previously published by the authors that first build a phylogenetic tree and then greedily insert edges into it (thereby turning it into a phylogenetic network) until all triplets are satisfied. As far as I can see, there are only two new contributions in this paper. The first one is to apply Wu's technique to select a good binary phylogenetic tree to use as the base, instead of just refining non-binary nodes to make make them binary in some arbitrary way. The second one is to consider every pair of existing edges when deciding the insertion point of each new edge in the last step, instead of inserting these new edges randomly. Both of these contributions are rather trivial adjustments. It seems like the authors are trying to publish a new paper every time they come up with some minor improvement to their own algorithm." 

Answer: As you know, the problem of constructing a phylogenetic network consistent with all given input triplets is NP-hard. The paper "Constructing the simplest possible phylogenetic network from triplets" firstly explained an exact exponential method to solve the problem [1]. Then based on the exact method, the heuristic SIMPLISTIC algorithm is introduced. Drawback of SIMPLISTIC is its disability to return a network for complex sets of triplets in an acceptable time. This scenario can occur when the number of taxa is growing, which yields increasing the complexity of relations among obtained triplets. More over this scenario happens to small sets of taxa with complex relation among triplets. 

We can conclude that working on phylogenetic networks needs considering more details about the assumptions, data size and the complexity of relations among input triplets. Therefore introducing an efficient method is an open challenge. 

By analyzing existing research we can divide the solution of constructing networks based on triplets, into two approaches. In the first approach, the reticulation nodes are recognized and then are removed from the set of taxa and a tree structure is obtained for the remaining taxa. Finally the network consistent with all given triplets is obtained by adding reticulation nodes to the tree structure. In the second approach, a tree structure is obtained and then by adding new edges to the tree structure, the final network consistent with all triplets is obtained. 

According to our best of knowledge, all the researches on this problem fall into one of these approaches. Therefore, in recent papers researchers try to improve these approaches gradually. It means that each improvement is valuable because it can reduce the time and costs, effectively. 

For example in MRTC problem which is simpler than network construction, the authors first proposed BPMF algorithm [2]. Then BPMR is introduced by improving BPMF [3]. Then by 

 example concerning with the first approach of constructing networks, TripNet algorithm was introduced [5]. Then NCHB was introduced that novelty improves TripNet [6]. 

According to the above explanations, we think that our proposed algorithm (NetCombin) which is an efficient algorithm in the second approach for constructing network is valuable and has appropriate novelty.

[1] Van Iersel Leo, Kelk Steven. Constructing the simplest possible phylogenetic network from triplets. Algorithmica. 2011;60(2):207-235.

[2] ‎Wu B.Y‎. ‎Constructing the maximum consensus tree from rooted triples. ‎Journal of‎ ‎Combinatorial Optimization‎. 2004; ‎8(1): ‎29-39‎‎. 

[3] ‎‎Maemura K‎, ‎‎Jansson J‎, Ono H‎, ‎‎Sadakane K‎, ‎‎Yamashita M. ‎Approximation algorithms‎ ‎for constructing evolutionary trees from rooted triplets.‎ ‎10th Korea-Japan joint‎ ‎workshop on algorithms and computation‎. ‎2007‎. 

[4] Jahangiri S, Hashemi SN, Poormohammadi H. New heuristics for rooted triplet consistency. Algorithms. 2013;6(3):396-406.

[5] Poormohammadi Hadi, Sardari Zarchi Mohsen. TripNet: a method for constructing rooted phylogenetic networks from rooted triplets. PloS one. 2014;6(6):e106531.

[6] Poormohammadi Hadi, Sardari Zarchi Mohsen, Ghaneai Hossein. NCHB: A Method for Constructing Rooted Phylogenetic Networks from Rooted Triplets based on Height Function and Binarization. Journal of Theoretical Biology. Volume 489, 21 March 2020, 110-144.

"The resulting method ("NetCombin") is not analyzed formally. This is no problem, though, since the authors evaluated its performance by comparing it experimentally against some other methods on simulated data. However, these comparisons do not seem fair because in the experiments, they applied NetCombin in parallel to construct 27 different networks and then output the best one found. For each of the other methods, they only constructed one network and output it. Thus, the running time of 44 seconds for NetCombin should be reported as 27*44=1188 seconds, etc., which means that it's actually slower than the NCHB method while outputting solutions that are worse than NCHB's. Furthermore, the experiments only go up to 40 sequences so it's difficult to draw any conclusions regarding the relative performance of NetCombin, TripNet, NCHB, and RPNCH."

Answer: We added a paragraph time complexity, formally in the section "Time complexity" of the manuscript. Experimentally we used a PC with Intel Corei7 CPU. We ran our algorithm on the cores of CPU in parallel. 

Concerning the simulated data, these triplet's data are obtained from biological data using standard methods. The triplet's data for Yeast are available [cite yeast]. The optimal network for yeast data is a level-2 network. Netcombin outputs for Yeast data, is a level-2 network which is optimal. There are no other biological standard triplets and its correspond optimal network. To obtain biological data we use standard methods to obtain triplets from generated biologically sequences data. This process is explained in details in subsection "Data generation" in "Experiments" section.

Usually existing methods are based on first approach (which explained in the first comment & answer). In this approach to reduce the size of the problem firstly SN-sets are recognized. Then a network is constructed for each SN-set. The network corresponds to a SN-set in the final network is the network that is connected to the final network via a cut edge. Note that each SN-set is a network itself. In the simple SN-sets there is no way to divide the problem into sub-problems. So generally if we consider simple SN-sets as inputs, the problem can not be solved in parallel. For example TripNet and NCHB and SIMPLISTIC belongs to the first approach that cannot become parallel. But our proposed method (NetCombin) which is based on the second approaches has the ability to solve in parallel process. We mentioned this fact in the revised version.

The previous methods like TripNet and NCHB uses at most 40 sequences for analyzing their performance. So in order to perform comparison we also used at most 40 sequences. However larger number of sequences can be used and our model can construct the network in an acceptable time.

"The authors don't even bother to explain all the steps of the method; in many places, they just refer to some of their old papers. It would have been better to list the complete algorithm. Throughout the paper. "

Answer: Thanks for your comments. As you suggested we added extra explanations to the revised manuscript about the steps of our algorithm. In the first part of method section, we explained the steps of NetCombin shortly. Also the part "constructing tree using height function" of the method section was divided into the two parts, "Assigning height function" and "obtaining tree". Moreover the method that we used to convert G_τ into a DAG was explained in details in the revised version. 

"There is lots of vague phrasing, mistakes, grammatical errors, and missing references to the literature.For example:

-The title is very strange.

The parameter k is never defined or used in the paper. 

What is k?

Does "optimal level-k network construction" mean that one wants to find a network that minimizes the value of k?"

Answer: Thanks for your careful attention. We review the manuscript to remove grammatical mistakes and typos errors. For example as you suggested:

- The title is changed to: "NetCombin: An algorithm for constructing optimal phylogenetic network from rooted triplets"

-There are two optimal criteria. First, minimizing the number of reticulation nodes and Second, minimizing the level of the network which is called K in this paper. A network is called level-k if the maximum number of reticulation nodes in each its bi-connected components is k. This explanation is added to the revised version. 

"Abstract and introduction:

It's better to say "or" instead of "and" when talking about minimizing the level and minimizing the number of reticulation nodes since they are not the same thing."

Answer: Logically your comment is right. Hence we used "or" instead of "and".

The abstract claims "The binarization process innovatively uses a measure to construct a binary rooted tree T consistent with the maximum number of input triplets.", but this is misleading.To find such a T is another NP-hard optimization problem and applying Wu's technique just gives an approximation of "a binary rooted tree consistent with the maximum number of input triplets", not always an optimal one. Similar phrasing occurs at the bottom of page 4.

Answer: Thanks for your careful comment. As you mentioned, the problem of constructing a rooted tree consistent with the maximum number of rooted triplets is an NP-hard problem. This problem is known as Maximum Rooted Triplets Consistency (MRTC). The goal is to find a acceptable (near the optimal) solution for MRTC. In order to remove this ambiguity we modify the paragraph based on your comment. Moreover we explain it in details in the introduction section. 

The abstract proudly declares that T is "expanded in an intellectual process".

Actually, it's just a straightforward greedy algorithm.

Answer: Although greedy algorithms seem to be straightforward, the most important things in them is defining how to define a heuristic function. One of our novelties is introducing an efficient heuristic function for constructing a phylogenetic network from obtained binary tree. However we changed the sentence to "Then ‎‎‎T‎ is expanded using a heuristic function by adding minimum number of edges to obtain final network with the approximately minimum number of reticulation nodes‎". 

The abstract says "The experimental results on real data indicate that". This is not true; only simulated data was used.

Answer: according to your valuable comment we adjust the sentence as follow: " The experimental results on simulated data obtained from biologically generated sequences data indicate that by considering the trade-off between speed and precision‎, ‎the NetCombin outperforms the others‎." 

We also modify it in experimental section as follow: "Secondly‎, ‎triplets can be obtained from sequences data‎. Sequences data usually are in the form of biological sequences‎. Biological sequences can be obtained from species or from simulation software that can generate these kinds of sequences under biological assumptions. In this research we used the second approach using simulation software."

"To make the problem definition easier to read, say clearly that all of the input triplets have to be consistent with the output network and that either the level or the number of reticulation nodes in the output network is to be minimized."

Answer: Thanks for your helpful comment. We add your suggested definition to the introduction part of the manuscript.

"When describing previous related results, it's important to explain how your old algorithms work since the new algorithm is a modification of them. My impression is that there is almost no difference, making this paper weak".

Answer: We explained novelty and differences between NetCombin and old methods in the answer of the first comment. If you suggest some parts of the answer of the first comment is better to be added to the manuscript please inform us. 

"The section "Definitions and notations" starts with the promise "In this section the basic definitions that are used in the proposed algorithm, are presented formally." A few sentences later, the text begins referring to something called "N", which has never been introduced."

Answer: In "Introduction" section we defined network N and in "Definitions and Notations" section we used it as N. However we modify definition by adding symbol N as follow: "‎Formally‎, ‎a rooted phylogenetic network ‎‎N‎‎ (network‎‎‎‎ for short) is a ...." 

"The "Method" section says "The height function enforce that the obtained rooted tree be consistent with maximum number of triplets of tau." In case no tree is consistent with all the triplets in tau, the IP won't have any valid solution, so I don't see how using the height function can yield a tree that is consistent with the maximum number of triplets in tau - How does NetCombin solve the IPs that were set up?"

Answer: We think you misunderstand the process. The height function finds an approximation solution for the IP which may ignore some triplets. If input triplets be consistent with a tree then the IP has a feasible solution. More precisely in this case the IP has the unique optimal solution. This optimal solution is corresponding to the height function related to the tree that is obtained by BUILD algorithm. If there is no tree consistent with a set of input triplets, then two cases happen. In the first case the IP has a feasible solution. In this case there is nothing to do and the algorithm goes to the next step. In the second case the IP has no feasible solution and we have an optimization (maximization) problem that should be solved heuristically. In this case the goal is to remove minimum information form the constraints of the IP. Corresponding to each triplet there are two constraints for the IP. The maximization problem is corresponding to Minimum Feedback Arc Set (MFAS) problem. Equivalently we assign a directed graph to the IP. The directed graph is acyclic if and only if the IP has a feasible solution. But in the second case the IP has no feasible solution and so the directed graph related to input triplets is not acyclic. To obtain a feasible solution for the IP we remove some constraints from it. It means that we solve the IP heuristically. Since the goal is to remove minimum number of information (constraints of the IP) so equivalently the goal is to remove minimum number of edges from the directed graph related to input triplets to make it acyclic. As mentioned this problem is MFAS which is an NP-hard problem. We used a heuristic method that is introduced in reference [14] of the manuscript. The resulting height function is used to obtain a tree by applying HBUILD algorithm. 

"What is the difference between BUILD and HBUILD? Line 207 suggests that they are "equivalent", but what does that mean?"

Answer: If the set of input triplets be consistent with a tree then the BUILD and HBUILD results are the same. Otherwise BUILD stops in some steps and gives no solution. But HBUILD can proceed consequently until it produces a tree. We mentioned this in the revised version. 

"In the experimental results, is the "number of sequences" the same thing as the "number of leaf labels"?"

Answer: Yes you are right. We added this tip in the revision.

"Lines 366-404 seem pointless. The reader can just look at the tables to get that information."

Answer: The result tables were presented concisely. To make it more understandable for the reader we explained them in more details in the manuscript. However if you think some sentences should be adjusted or removed please let us to know your suggestions.

"Does Table 4 list the number of reticulations or the level? The caption says one thing but the table itself says another thing."

Answer: Thanks for your careful attention. We corrected table 4.

"The example in Figure 5 is wrong. The input set of triplets contains a triplet de|f, which means that the BUILD algorithm will join d and e by an edge so that all of the leaf labels end up in the same connected component."

Answer: Thanks for your comment. As figure 5 shows, in the input triplets of tau, de|f should be da|f. We corrected this in the manuscript. 

"The caption of Figure 5 is misleading. It says "two nodes i,j in X are connected iff" but usually two nodes are said to be "connected" if there is a path (of any length) between them.

It would be better to say "two nodes i,j in X are adjacent iff" or "two nodes i,j in X are connected by an edge iff". "

Answer: Thanks for your helpful comment. We adjust the caption of figure 5, by replacing ''connected'' with ''adjacent''.

"The caption of Figure 5 refers to the "Aho graph". However, the BUILD algorithm was published in a paper written by four people. To be fair, call it "the Aho-Sagiv-Szymanski-Ullman graph" or something like that if you insist on using family names."

Answer: You are right. However it is very common to use "Aho graph" term. For example in the book of reference [2] of the manuscript (which is a main book for Phylogenetics) the term "Aho graph" is used frequently.

"The example in Figure 10 is wrong. The two trees in (b) and (c) are not binary.

In both trees, the node lca(f,g) has three children. Also, the text refers to "its two different binarizations" but there are many more possible binarizations."

Answer: Thanks for your critical point. We corrected figure 10. Additionally we mentioned that there are more than binarizations in the text. 

"The caption of Figure 14 uses Max-Cut in part (m). Should it be Min-Cut?"

Answer: In part (l) of figure 14 Min-Cut is used and in part (m) Max-Cut is used. 

"Lines 321-322 state some strange-looking time complexity (without any

proof) for one of the steps in the algorithm. It doesn't seem relevant as the time complexity of all the other steps has been completely ignored."

Answer: Thanks for your comment. As you suggested we added the part "Time complexity" to the revised version to explain the complexity of NetCombin in more details.

"More than 30% of the references in the bibliography are self-references. This is too much. On the other hand, many historically important bibliographic references are missing, such as the following."

Answer: As you know this research is based on our previous works. So we have to cite them. However as you suggested we cited new references you mentioned in the comment.

Reviewer 2:

"1.) The authors e.g., in the paragraph from lines 10-14 discuss the importance of constructing graphs instead of trees. However, the first example are just trees. Further, the terminology is not consistent. Even though each tree is a graph, not each graph is a tree, thus the authors should stick to notation of directed graphs in my opinion. The comment is related to the following text:"

“The rooted structures 12 are directed graphs which contains a unique vertex called root with in-degree 0 and 13 out-degree at least 2 .. Figure 1a shows an example of a rooted tree.”

Answer: Thanks for your comment. We changed the sentences to "The rooted structures are always rooted trees or rooted networks. These structures contain a unique vertex called root with in-degree 0 and out-degree at least two. "

"2.) The quality of the second figure is very poor and cut off. Please correct this."

Answer: We adjusted the second figure.

"3.) line 48: please define the notion of the level of a network. Even though this is a basic concept, it appears here for the first time without any clarification."

Answer: As you suggested we brought the definition of the level of a network before the sixth paragraph of the introduction. 

"4.) Line 64: So, introducing efficient heuristic methods to solve this problem is demanding. Perhaps “necessary” instead of “demanding” is meant?"

Answer: Thanks for your point. We replaced it. 

"5.) The name of the proposed algorithm is NetCombin, however this is not consistent throughout the paper, please unify."

Answer: The name NetCombin stands for Network construction method based on binarization. Based on your comment we unified the name as Netcombin across the manuscript.

"6.) Line 160. Is N the set of natural numbers with 0 or without?"

Answer: N is the set of natural numbers without 0.

Comments related to the method:

"1.) Line 218, please discuss in more detail on MFAS heuristic used. This is highly relevant to the proposed approach."

Answer: We added more details about MFAS problem and the heuristic method that was used to solve it in subsection "assigning height function".

"2.) Line 230, what are the three “???”? Explain."

 Answer: It was redundant and we removed it. 

"3.) Line 282 onwards a couple of lines. The authors provide the pseudocode of the proposed algorithm, however, it is not clear why all 9 different measures for construction of binary trees (as apparently used in [5]). This computational step appears somewhat redundant and seems like everything that can be, is computed. In line 301, authors claim innovative use. Could this be elaborated in more detail? I don’t see how exhaustive enumeration of arbitrary 9 measures is any more innovative than doing this for more measures, unless the 9 offer some form of theoretical grounding that makes this traversal/scoring feasible? Furthermore, how can one resolve possible ties? What if the score distribution is ~uniform? Is the subtree selected at random? Please elaborate on these details."

Answer: The 9 measures are equivalent to 9 heuristic functions which give better output compared to other heuristic functions. In defining these 9 heuristic functions, we used the parameters W, P and T which are the important factors in tree construction methods based on rooted triplets. These parameters were introduced by Wuo …..

The subtrees are selected based on SPR method and were described in details in the manuscript. The term uniform if your mean is that the best tree uniformly distributed between these 9 functions, the answer is no. 

"4.) Line 309 onwards. It seems that the network construction step is rather expensive (n^3, m^2). To what extent can this be made run in parallel. Could you discuss this aspect, at least theoretically?"

Answer: We added the new section "time complexity" and described the time complexity in details. In the previous submitted version, we just mention the time complexity of tree construction. In the revised version time complexity was explained for all parts of Netcombin rather than only tree construction step.

"I don’t quite understand the reasoning behind: “but NetCombin 360 outputs 27 networks and the best network is reported. Since the process of constructing 361 these 27 NetCombin networks is indepe”

Why is only the best one reported? Isn’t this somewhat non-representative for real-life situations. Could you elaborate on that? Further, do other methods also output multiple networks? How did you select the compared against there?

Further, if you report the average running time, please repreat the experiments at least 5 times and report also the standard deviations. "

Answer: we analyzed problem based on the two optimality criterions and best network result is reported. Sine our method is based on optimizing the number of reticulation nodes, in almost all cases optimal network is unique. 

Biologically some results may be more informative compared to optimal result. We did not consider it biologically since our data are semi real and are generated based on biological constraints. We ran our algorithm in parallel on cores of Cori7 CPU and the time all process was considered. There is no average computing in our experiment and so we do not have standard deviations parameter. The resulting network of NCHB, TripNet, SIMPLISTIC and RPNCH is unique.

"I am not entirely sure what the measure of success is in T - it shows the number of outputted networks? Is more better? Why?

Further, low runtime is meaningless if the quality of results is low. Comment on the runtime w.r.t T3 and T4."

Answer: As mentioned before the problem of constructing an optimal network consistent with all given input triplets is NP-hard and different methods are designed for solving the problem, heuristically. In order to study the performance of Netcobmin we generated data sets and compared our method with TripNet, SIMPLISTIC, NCHB and RPNCH. The comparison is based on three parameters level, number of reticulation nodes and running time. By considering all these three parameters the results show that our method outperforms other methods. For example RPNC is very fast but its results optimality is very low. SIMPLISTIC results for small size data, is good. But by increasing the number of taxa to 20 and more, SIMPLISTIC runtime exceeds exponentially and for large size data is not applicable. NCHB and TripNet works well for small size data. By increasing the number of input taxa the running time of both methods exceeds. Note that on average the performance of TripNet and NCHB compared to SIMPLISTIC is better with respect to running time and optimality criterions. For large size data and by increasing the number of taxa, the running time of TripNet and NCHB increased very faster compared to Netcombin. For large size data the performance of Netcombin is nearly the same as NCHB and TripNet, but its running time is very low. It means that by considering all three parameters, on average Netcombin is a reasonable method for solving the problem. So both low runtime and low level and low number of reticulation nodes happens simultaneously for large size data for Netcombin results. Netcombin firstly constructs 27 trees and then by adding edges to these obtained trees, it constructs 27 networks and optimal network is reported as output. In tree construction process and adding edges, approximately optimal trees and networks are constructed innovatively. Firstly approximately optimal trees are constructed innovatively by using 9 measures and using SPR with respect to the consistency with the maximum number of input triplets. Then approximately optimal network is constructed from each tree and by adding some edges in an innovative way to reduce the number of added reticulation nodes. 

"There is not enough discussion on the results (see previous section). Further, the last line reads as “NetCombin is a 434 valuable method that returns an appropriate network in an appropriate time”. What does that mean? What is appropriate? Do you mean reasonable?

Why would the user prefer NetCombin, if it performs on par with existing state-of-the-art?

Please comment on this issue.

The authors should further discuss why NCHB is a worse alternative, as it seems to perform very similarly. Is the runtime main benefit of NetCombin?"

Answer: TripNet has three speed options slow, normal, and fast. The results of slow speed option are better compared to normal and normal speed option results are better compared to fast speed option. In almost all cases the normal speed option is used. By increasing the number of taxa, using slow or normal speed options are time consuming. In almost all cases and for comparing TripNet with other methods, normal speed option is used. In this manuscript TripNet results are based on normal speed option.

We added some sentences to the beginning of discussion part to explain our method in more details. Also thanks for your comment and we replaced the word appropriate with reasonable.

As we mentioned in the previous comments, for large size data and by increasing the number of taxa, Netcombin output a network in a time very better compared to NCHB while its optimality is nearly the same as NCHB. For small size data one can use each of Netcombin, TripNet or NHCB. But for large size data which are very common in real data, Netcombin is suggested by considering running time and optimality criterions.

"Statistical evaluation:

The authors propose tabelaric comparison of results, however, such results do not offer statistically sound insights into the algorithm's inner workings. I would suggest the authors to consider at least critical distance diagrams if possible."

Answer: We did not consider the comparison statistically. Previous methods in constructing trees and networks based on input triplets, were not analyzed statistically and just their performance are compared based on the running time and optimality criterions. If the reviewer suggests a new statistical evaluation, please inform us.

"Availability:

Please, make the code and data simulators (or datasets) publicly available via .e.g, github."

Answer: Netcobimn is not public and is partially supported under a grant. Some parts of the method can be reported publicly or can be sent to the reviewer. We sent the part related to tree construction . 

Final remarks:

All images are of terrible quality. Please, render the plots as vector graphics if possible, otherwise use >400 dpi. Further, the authors should specify what exactly are the main adopted novelties at the end of the introductory section.

Answer: All images are designed in visio and then are converted to JPG format. Visio is vector based and JPG is not. We can send the original visio source. Also we used >400 dpi for all figures in the revised version. At the end of introduction section we briefly mentioned the novelties of our method. In the new version we added some sentences to the discussion part to show the novelties and importance of our new method.

---

## [Decision Letter · Decision Letter 1]

27 Aug 2020

PONE-D-19-34247R1

Netcombin‎: ‎An algorithm for constructing optimal phylogenetic network from rooted triplets

PLOS ONE

Dear Dr. Poormohammadi,

Thank you for submitting your manuscript to PLOS ONE. After careful consideration, we feel that it has merit but does not fully meet PLOS ONE’s publication criteria as it currently stands. Therefore, we invite you to submit a revised version of the manuscript that addresses the points raised during the review process.

We look forward to receiving your revised manuscript.

Kind regards,

Hocine Cherifi

Academic Editor

PLOS ONE

Additional Editor Comments (if provided):

Please update the figure

Reviewers' comments:

Reviewer's Responses to Questions

**Comments to the Author**

1. If the authors have adequately addressed your comments raised in a previous round of review and you feel that this manuscript is now acceptable for publication, you may indicate that here to bypass the “Comments to the Author” section, enter your conflict of interest statement in the “Confidential to Editor” section, and submit your "Accept" recommendation.

Reviewer #2: All comments have been addressed

2. Is the manuscript technically sound, and do the data support the conclusions?

Reviewer #2: Yes

3. Has the statistical analysis been performed appropriately and rigorously? 

Reviewer #2: Yes

4. Have the authors made all data underlying the findings in their manuscript fully available?

Reviewer #2: Yes

5. Is the manuscript presented in an intelligible fashion and written in standard English?

Reviewer #2: Yes

6. Review Comments to the Author

Reviewer #2: The authors have addressed the most of my concerns. The only thing still bothering me is the low resolution of the figure fig14e.jpg (I'm not sure this is >400 dpi).

I'd suggest you improve the quality prior to publication.

7. PLOS authors have the option to publish the peer review history of their article (what does this mean?). If published, this will include your full peer review and any attached files.

Reviewer #2: No

---

## [Editor Report · Decision Letter 2]

9 Sep 2020

Netcombin‎: ‎An algorithm for constructing optimal phylogenetic network from rooted triplets

PONE-D-19-34247R2

Dear Dr. Poormohammadi,

We’re pleased to inform you that your manuscript has been judged scientifically suitable for publication and will be formally accepted for publication once it meets all outstanding technical requirements.

Kind regards,

Hocine Cherifi

Academic Editor

PLOS ONE
---

## [Editor Report · Acceptance letter]

11 Sep 2020

PONE-D-19-34247R2 

Netcombin‎: ‎An algorithm for constructing optimal phylogenetic network from rooted triplets 

Dear Dr. Poormohammadi:

I'm pleased to inform you that your manuscript has been deemed suitable for publication in PLOS ONE. Congratulations! Your manuscript is now with our production department. 

Kind regards, 

on behalf of

Professor Hocine Cherifi 

Academic Editor

PLOS ONE